# Auditory confounds can drive online effects of transcranial ultrasonic stimulation in humans

**Benjamin R Kop**[1]*, **Yazan Shamli Oghli**[2]†, **Talyta C Grippe**[2]†, **Tulika Nandi**[3]†, **Judith Lefkes**[1], **Sjoerd W Meijer**[1], **Soha Farboud**[1], **Marwan Engels**[1], **Michelle Hamani**[2], **Melissa Null**[3], **Angela Radetz**[3], **Umair Hassan**[3], **Ghazaleh Darmani**[2], **Andrey Chetverikov**[1,4], **Hanneke EM den Ouden**[1]‡, **Til Ole Bergmann**[3,5]‡, **Robert Chen**[2]‡, **Lennart Verhagen**[1]

[1]Donders Institute for Brain, Cognition, and Behaviour; Radboud University Nijmegen, Nijmegen, Netherlands; [2]Krembil Research Institute, University Health Network; University of Toronto, Toronto, Canada; [3]Neuroimaging Center; Johannes-Gutenberg University Medical Center Mainz, Mainz, Germany; [4]Department of Psychosocial Science, Faculty of Psychology, University of Bergen, Bergen, Norway; [5]Leibniz Institute for Resilience Research Mainz, Mainz, Germany

**\*For correspondence:**
benjamin.kop@donders.ru.nl

†These authors contributed equally to this work
‡These authors also contributed equally to this work

## eLife assessment

This **important** multicenter study provides **convincing** evidence that the auditory noise emitted during online transcranial ultrasound stimulation (TUS) protocols can pose a considerable confound and is able to explain corticospinal excitability changes as measured with Motor Evoked Potentials (MEP). The findings lay the ground for future studies optimising protocols and control conditions to leverage TUS as a meaningful experimental and clinical tool. A clear strength of the study is the multitude of control conditions (i.e., control sites, acoustic masking, acoustic stimulation). These findings will be of interest to neuroscience researchers using brain stimulation approaches.

**Abstract** Transcranial ultrasonic stimulation (TUS) is rapidly emerging as a promising non-invasive neuromodulation technique. TUS is already well-established in animal models, providing foundations to now optimize neuromodulatory efficacy for human applications. Across multiple studies, one promising protocol, pulsed at 1000 Hz, has consistently resulted in motor cortical inhibition in humans (Fomenko et al., 2020). At the same time, a parallel research line has highlighted the potentially confounding influence of peripheral auditory stimulation arising from TUS pulsing at audible frequencies. In this study, we disentangle direct neuromodulatory and indirect auditory contributions to motor inhibitory effects of TUS. To this end, we include tightly matched control conditions across four experiments, one preregistered, conducted independently at three institutions. We employed a combined transcranial ultrasonic and magnetic stimulation paradigm, where TMS-elicited motor-evoked potentials (MEPs) served as an index of corticospinal excitability. First, we replicated motor inhibitory effects of TUS but showed through both tight controls and manipulation of stimulation intensity, duration, and auditory masking conditions that this inhibition was driven by peripheral auditory stimulation, not direct neuromodulation. Furthermore, we consider neuromodulation beyond driving overall excitation/inhibition and show preliminary evidence of how TUS might interact with ongoing neural dynamics instead. Primarily, this study highlights the substantial shortcomings in accounting for the auditory confound in prior TUS-TMS work where only a flip-over sham and no active control was used. The field must critically reevaluate previous findings given the

demonstrated impact of peripheral confounds. Furthermore, rigorous experimental design via (in) active control conditions is required to make substantiated claims in future TUS studies. Only when direct effects are disentangled from those driven by peripheral confounds can TUS fully realize its potential for research and clinical applications.

## Introduction

Noninvasive neuromodulation is a powerful tool for causal inference that strengthens our understanding of the brain and holds great clinical potential (*Bergmann and Hartwigsen, 2021*; *Bestmann and Walsh, 2017*). Transcranial ultrasonic stimulation (TUS) is a particularly promising non-invasive brain stimulation technique, overcoming current limitations with high spatial resolution and depth range (*Darmani et al., 2022*). The efficacy of TUS is well-established in cell cultures and animal models (*Menz et al., 2013*; *Mohammadjavadi et al., 2019*; *Murphy et al., 2022*; *Tyler et al., 2008*; *Tyler et al., 2018*; *Yoo et al., 2022*), and emerging evidence for the neuromodulatory utility of TUS in humans has been reported for both cortical and subcortical structures (cortical: *Butler et al., 2022*; *Lee et al., 2016*; *Liu et al., 2021*; *Zeng et al., 2022*; subcortical: *Ai et al., 2016*; *Cain et al., 2021*; *Nakajima et al., 2022*). Especially now, at this foundational stage of TUS in humans, it is essential to converge on protocols that maximize the specificity and efficacy of stimulation (*Folloni et al., 2019*; *Verhagen et al., 2019*).

Motor inhibitory effects of a commonly applied 1000 Hz pulsed TUS protocol are among the most robust and replicable human findings (*Fomenko et al., 2020*; *Legon et al., 2018b*; *Xia et al., 2021*). Here, by concurrently applying transcranial magnetic stimulation (TMS), modulation of corticospinal excitability is indexed by motor-evoked potentials (MEPs). However, the mechanism by which TUS evokes motor inhibition has remained under debate (*Xia et al., 2021*).

Recent studies in both animal and human models demonstrate how electrophysiological and behavioral outcomes of TUS can be elicited by nonspecific auditory activation rather than direct neuromodulation (*Airan and Butts Pauly, 2018*; *Braun et al., 2020*; *Guo et al., 2018*; *Sato et al., 2018*). Indeed, there is longstanding knowledge of the auditory confound accompanying pulsed TUS (*Gavrilov and Tsirulnikov, 2012*). However, this confound has only recently garnered attention, prompted by a pair of rodent studies demonstrating indirect auditory activation induced by TUS (*Guo et al., 2022*; *Sato et al., 2018*). Similar effects have been observed in humans, where exclusively auditory effects were captured with EEG measures (*Braun et al., 2020*). These findings are particularly impactful given that nearly all TUS studies employ pulsed protocols, from which the pervasive auditory confound emerges (*Johnstone et al., 2021*).

Indirect effects of stimulation are not unique to TUS, as transcranial magnetic and electric stimulation are also associated with auditory and somatosensory confounds. Indeed, the field of non-invasive brain stimulation as a whole depends on controlling for these confounding factors when present, to unveil the specificity of the neuromodulatory effects (*Conde et al., 2019*; *Duecker et al., 2013*; *Polanía et al., 2018*; *Siebner et al., 2022*). However, prior online TUS-TMS studies, including those exploring optimal neuromodulatory parameters to inform future work, have considered some but not all necessary conditions to control for the salient auditory confound elicited by a 1000 Hz pulsed protocol (*Fomenko et al., 2020*; *Legon et al., 2018b*; *Xia et al., 2021*).

In this multicenter study, we quantified the impact of the auditory confound to disentangle direct neuromodulatory and indirect auditory contributions to motor inhibitory effects of TUS. To this end, we substantially improved upon prior TUS-TMS studies implementing solely flip-over sham by including both (in)active controls and multiple sound-sham conditions. Furthermore, we investigated dose-response effects through the administration of multiple stimulus durations, stimulation intensities, and individualized simulations of intracranial intensity. Additionally, we considered the possibility that online TUS might not drive a global change in the excitation/inhibition balance but instead might interact with ongoing neural dynamics by introducing state-dependent noise. Finally, we interrogated sound-driven effects through modulation of auditory confound volume, duration, pitch, and auditory masking. We show that motor inhibitory effects of TUS are spatially nonspecific and driven by sound-cued preparatory motor inhibition. However, we do find preliminary evidence that TUS might introduce dose- and state-dependent neural noise to the dynamics of corticospinal excitability. The present study highlights the importance of carefully constructed control conditions to

account for confounding factors while exploring and refining TUS as a promising technique for human neuromodulation.

## Results

### Motor cortical inhibition is not specific to on-target TUS

We first corroborate previous reports of MEP suppression following 500ms of TUS applied over the hand motor area (Experiments I-III; *Fomenko et al., 2020*; *Legon et al., 2018b*; *Xia et al., 2021*). A LMM revealed significantly lower MEP amplitudes following on-target TUS as compared to baseline for Experiment I ($b$=–0.14, SE = 0.06, $t$(11) = –2.23, p=0.047), Experiment II ($b$=–0.18, SE = 0.04, $t$(26) = –4.82, p=$6 \cdot 10^{-5}$), and Experiment III ($b$=–0.22, SE = 0.07, $t$(15) = –3.08, p=0.008).

However, corticospinal inhibition from baseline was also observed following control conditions. LMMs revealed significant attenuation of MEP amplitude following active control stimulation of the right-hemispheric face motor area (Experiment I: $b$=–0.12, SE = 0.05, $t$(11) = –2.29, p=0.043; Experiment II: $b$=–0.22, SE = 0.04, $t$(26) = –5.60, p=$7 \cdot 10^{-6}$), as well as after inactive control stimulation of the white matter ventromedial to the left-hemispheric hand motor area (Experiment IV: $b$=–0.14; SE = 0.04; $t$(11) = –3.09; p=0.010). The same effect was observed following sound-only sham (Experiment I: $b$=–0.14; SE = 0.05; $t$(11) = –3.18; p=0.009; Experiment II: $b$=–0.22; SE = 0.04; $t$(26) = –5.38; p=$1 \cdot 10^{-5}$; Experiment III: 500 ms-1kHz; $b$=–0.24; SE = 0.08; $t$(15) = –2.86; p=0.012). These results suggest a spatially non-specific effect of TUS that is related to the auditory confound (*Figure 1*).

### No dose-response effects of TUS on corticospinal inhibition

We further tested for direct ultrasonic neuromodulation by investigating a potential dose-response effect of TUS intensity ($I_{sppa}$) on motor cortical excitability. First, we applied TUS at multiple free-water stimulation intensities (*Figure 2C*). In Experiment I, a linear mixed model with the factor 'intensity' (32.5/65 W/cm$^2$) did not reveal a significant effect of different on-target TUS intensities on motor excitability ($F$(1,11) = 0.47, p=0.509, $\eta_p^2$ = 0.04). In Experiment II, a linear mixed model with the factors 'stimulation site' (on-target/active control), 'masking' (no mask/masked), and 'intensity' (6.35/19.06 W/cm$^2$) similarly did not reveal an effect of stimulation intensity ($F$(1,50) = 1.29, p=0.261, $\eta_p^2$ = 0.03). Importantly, there was no effect of stimulation site ($F$(1,168) = 1.75, p=0.188, $\eta_p^2$ = 0.01), nor any significant interactions (all p-values >0.1; all $\eta_p^2$ < 0.06). These results provide neither evidence for spatially specific neuromodulation when directly comparing stimulation sites, nor evidence for a dose-response relationship within the range of applied intensities.

However, it is likely that the effectiveness of TUS depends primarily on realized intracranial intensity, which we estimated with individualized 3D simulations (*Figure 2A*). Yet, testing the relationship between estimated intracranial intensity and MEP amplitude change following on-target TUS similarly did not yield evidence for a dose-response effect (*Figure 2B*, Appendix 1).

Prior work has primarily focused on probing facilitatory or inhibitory effects on corticospinal excitability. Here, we also explored an alternative: how TUS might introduce noise to ongoing neural dynamics, rather than a directional modulation of excitability. Indeed, human TUS studies have often failed to show a global change in behavioral performance, instead finding TUS effects primarily around the perception threshold where noise might drive stochastic resonance (*Butler et al., 2022*; *Legon et al., 2018a*). Whether the precise principles of stochastic resonance generalize from the perceptual domain to the current study is an open question, but it is known that neural noise can be introduced by brain stimulation (*van der Groen and Wenderoth, 2016*). It is likely that this noise is state-dependent and might not exceed the dynamic range of the intra-subject variability (*Silvanto et al., 2007*). Therefore, in an exploratory analysis, we exploited the natural structure in corticospinal excitability that exhibits as a strong temporal autocorrelation in MEP amplitude. Specifically, we tested how strongly the MEP on test trial *t* is predicted by the previous baseline trial *t-1*. As such, we quantified state-dependent autocorrelation between baseline MEP amplitude and MEP amplitude following on-target TUS, active control TUS, and sound-sham conditions (Appendix 2). In brief, we found a significant interaction between the previous baseline (*t-1*), stimulation site (on-target/active control), and intensity (6.35/19.06 W/cm$^2$; $F$(1,30) = 12.10, p=0.002, $\eta_p^2$ = 0.28) during masked trials. This interaction exhibited as increased autocorrelation for on-target TUS compared to active control TUS at 6.35 W/cm$^2$ (i.e. lower TUS-induced noise; $F$(1,1287)=13.43, p=$3 \cdot 10^{-4}$, $\eta_p^2$ = 0.01), and reduced

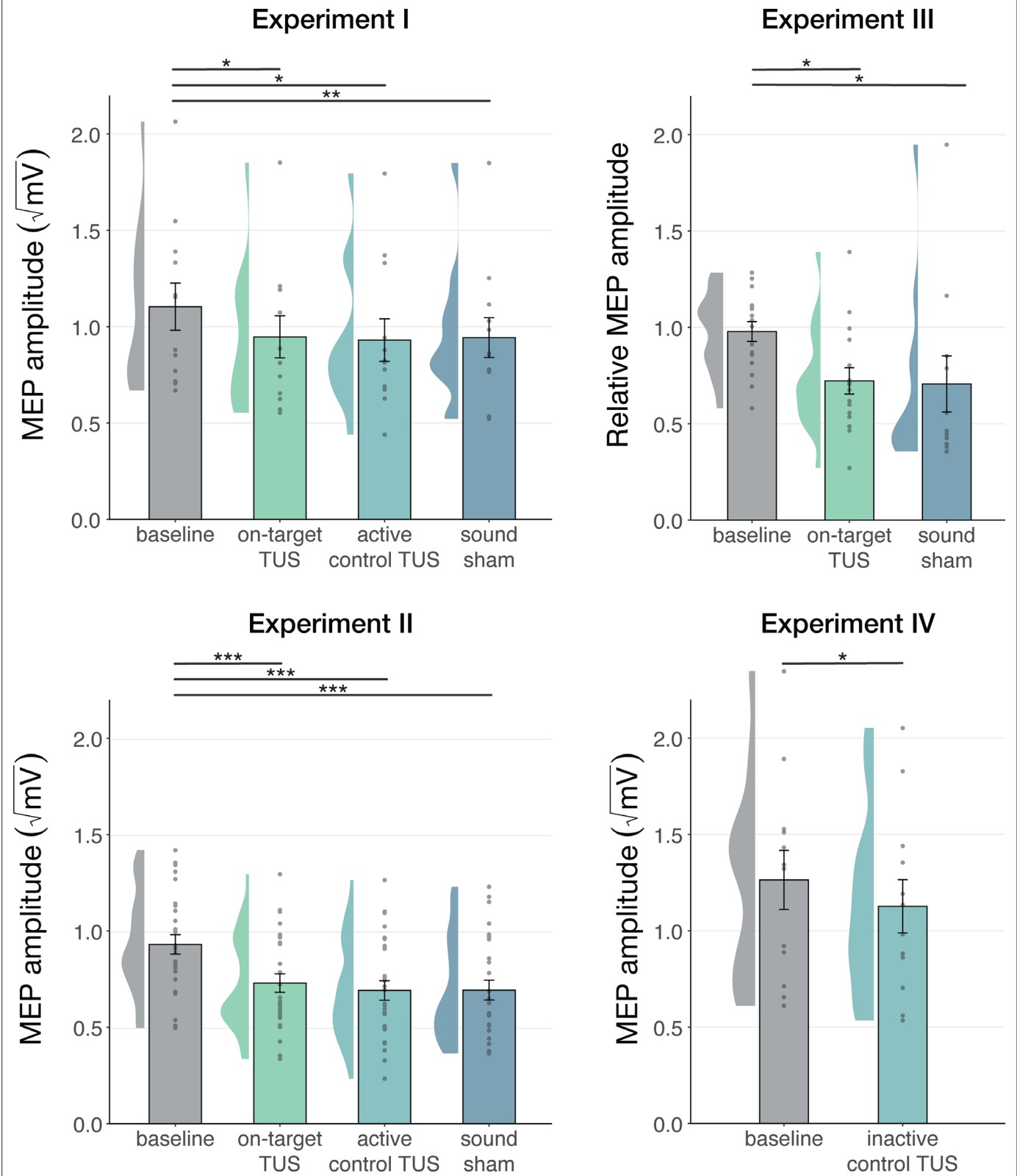

**Figure 1.** Non-specific motor inhibitory effects of transcranial ultrasonic stimulation (TUS). A significant suppression of motor-evoked potentials (MEP) amplitude relative to baseline (gray) was observed for on-target TUS (green), but also for stimulation of a control region (cyan), and presentation of a sound alone (sound-sham; blue) indicating a spatially non-specific and sound-driven effect on motor cortical excitability. There were no significant differences between on-target and control conditions. Bar plots depict condition means, error bars represent standard errors, clouds indicate the

*Figure 1 continued on next page*

*Figure 1 continued*

distribution over participants, and points indicate individual participants. Square-root corrected MEP amplitudes are depicted for Experiments I, II, and IV, and Relative MEP amplitude is depicted for Experiment III (see Methods). *p<0.05, **p<0.01, ***p<0.001.

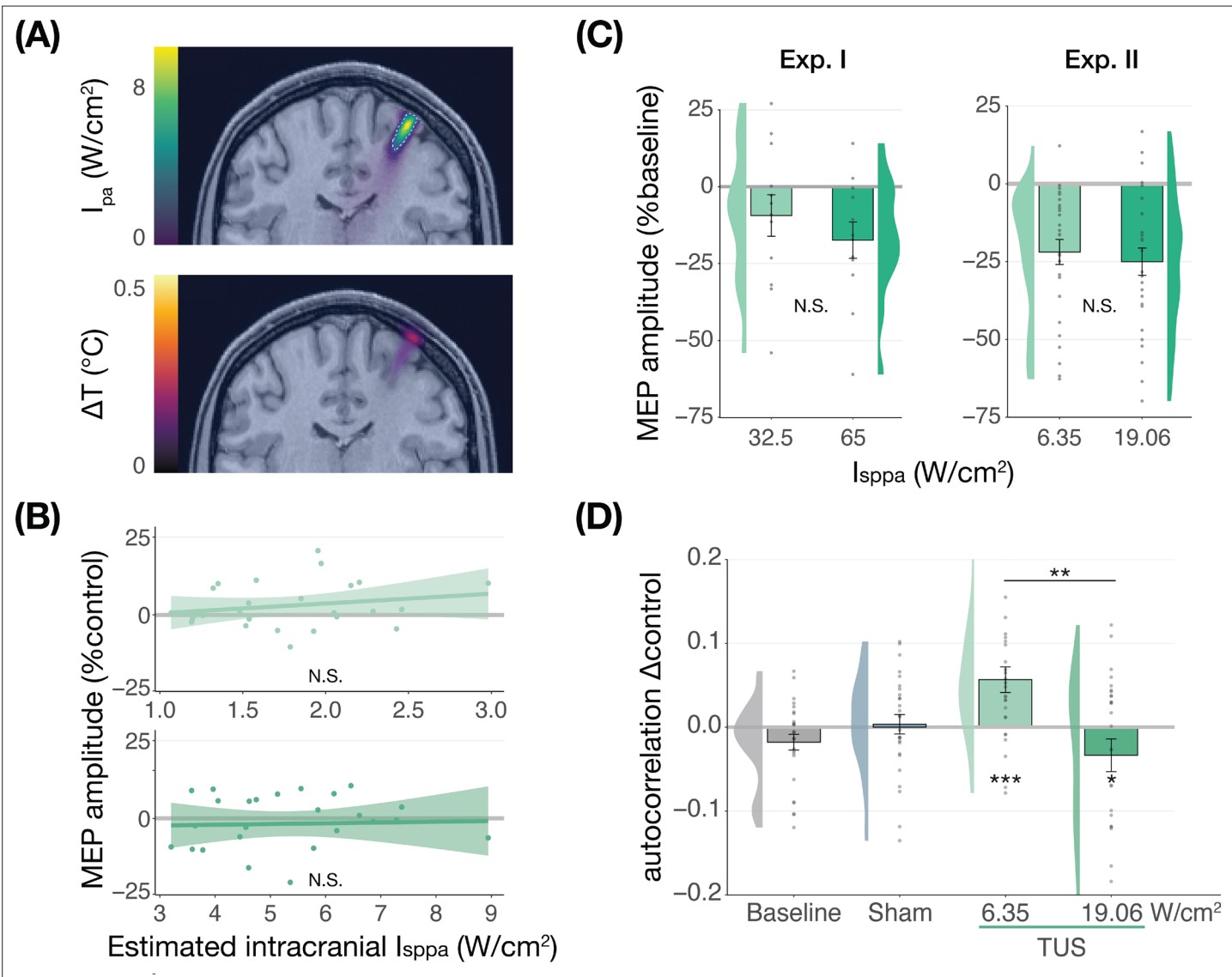

**Figure 2.** No significant dose-response effects of transcranial ultrasonic stimulation (TUS). (**A**) Acoustic (top) and thermal (bottom) simulations for a single subject in Experiment II. The acoustic simulation depicts the estimated pulse-average intensity ($I_{pa}$) above a 0.15 W/cm² lower bound, with the dotted line indicating the full-width half-maximum of the pressure. The thermal simulation depicts the maximum estimated temperature rise. See Appendix 5 for more information on simulations . (**B**) On-target TUS motor-evoked potentials (MEP) amplitude as a percentage of active control MEP amplitude against simulated intracranial intensities at the two applied free-water intensities: 6.35 W/cm² (top) and 19.06 W/cm² (bottom). The shaded area represents the 95% CI, points represent individual participants. No significant intracranial dose-response relationship was observed. (**C**) There is no significant effect of free-water stimulation intensity on MEP amplitude. Values are expressed as a percentage of baseline MEP amplitude (square root corrected). Remaining conventions are as in *Figure 1*. (**D**) Temporal autocorrelation, operationalized as the slope of the linear regression between trial *t* and its preceding baseline trial *t*-1, differed significantly as a function of stimulation site and intensity for masked trials. Individual points represent the differential autocorrelation compared to the active control site. Autocorrelation was not modulated during baseline or sound-only sham, but was significantly higher for on-target TUS at 6.35 W/cm², and significantly lower for on-target TUS at 19.06 W/cm² compared to active control TUS. *p<0.05, **p<0.01, ***p<0.001.

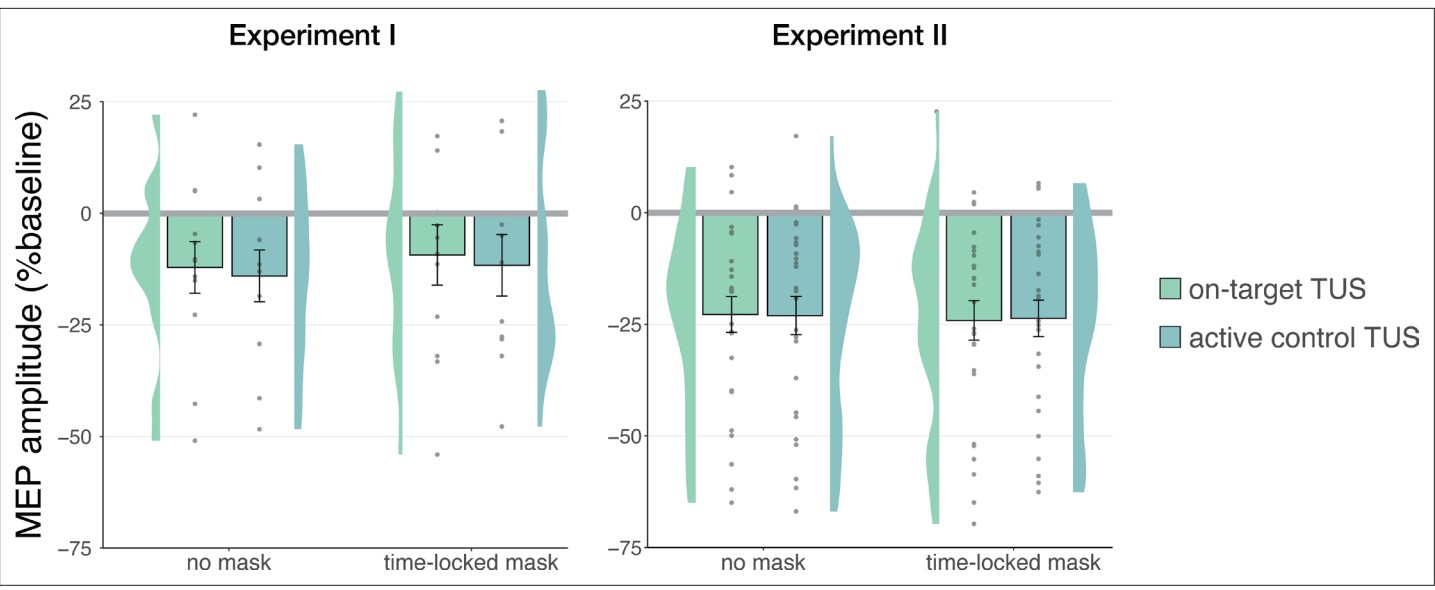

**Figure 3.** No effects of time-locked masking. There were no significant effects of time-locked masking, indicating that audible differences between stimulation sites did not obscure or explain the absence of direct neuromodulation. Conventions are as in *Figures 1 and 2*.

autocorrelation at 19.06 W/cm$^2$ (i.e. higher noise; $F(1,1282)=5.76$, p=0.017, $\eta_p^2 = 4\cdot10^{-3}$; *Figure 2D*). This effect was not only dependent upon intensity and stimulation site, but also dependent on the presence of auditory masking. As such, the effect was also observed in a four-way interaction of the previous baseline, site, intensity, and masking (Appendix 2). These preliminary results might suggest that ultrasound stimulation can interact with ongoing neural dynamics by introducing temporally specific noise, rather than biasing the overall excitation/inhibition balance beyond its natural variation, but further work specifically designed to detect such effects is required.

### Audible differences between stimulation sites do not underlie nonspecific inhibition

Stimulation over two separate sites could evoke distinct perceptual experiences arising from bone-conducted sound (*Braun et al., 2020*). To account for possible audibility differences between stimulation of on-target and active control sites in Experiments I and II, we also tested these conditions in the presence of a time-locked masking stimulus (*Figure 3*). Following Experiment II, we additionally assessed the blinding efficacy of our masking stimuli, finding that the masking stimulus effectively reduced participant's the ability to determine whether TUS was administered to approximately chance level (Appendix 3).

In Experiment I, a linear mixed model with factors 'masking' (no mask/masked) and 'stimulation site' (on-target/active control) did not reveal a significant effect of masking ($F(1,11) = 0.01$, p=0.920, $\eta_p^2 = 1\cdot10^{-5}$), stimulation site ($F(1,11) = 0.15$, p=0.703, $\eta_p^2 = 0.01$), or their interaction ($F(1,11) = 1\cdot10^{-3}$, p=0.971, $\eta_p^2 = 1\cdot10^{-4}$). Similarly, in Experiment II, the linear mixed model described in the previous section revealed no significant main effect of masking ($F(1,30) = 1.68$, p=0.205, $\eta_p^2 = 0.05$), nor any interactions (all p-values >0.1; all $\eta_p^2 < 0.06$). These results indicate that an underlying specific neuro-modulatory effect of TUS was not being obscured by audible differences between stimulation sites.

### Sound-driven effects on corticospinal excitability

#### Duration and pitch

Prior research has shown that longer durations of TUS significantly inhibited motor cortical excitability (i.e. ≥400 ms; *Fomenko et al., 2020*), while shorter durations did not. In Experiment I, we applied on-target, active control, and sound-sham conditions at shorter and longer durations to probe this effect. When directly comparing these conditions at different stimulus durations (100/500 ms), no evidence for an underlying neuromodulatory effect of TUS was observed, in line with our afore-mentioned findings. Instead, a linear mixed model with factors 'condition' (on-target/active control/

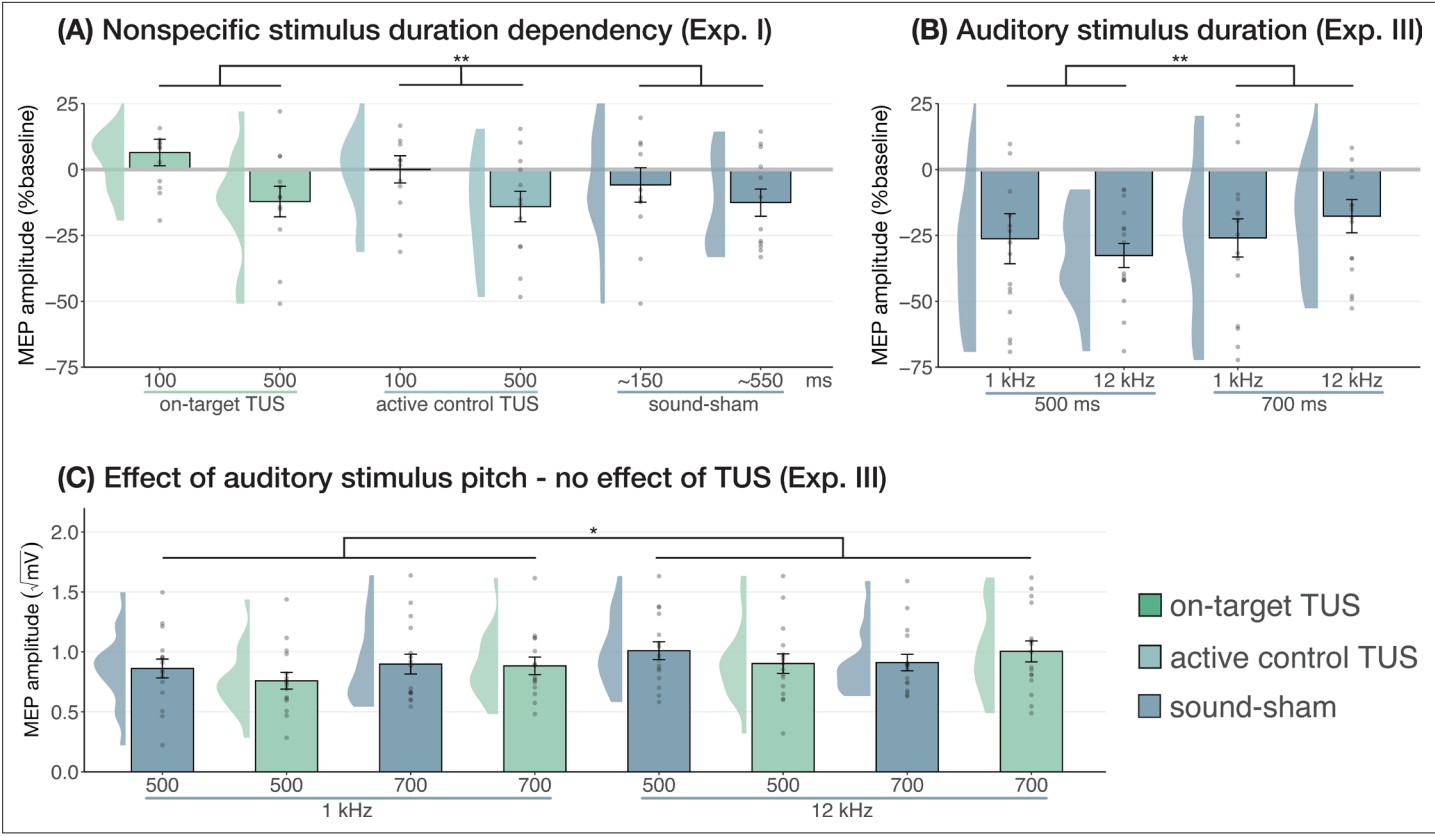

**Figure 4.** Sound-driven effects on corticospinal excitability. (**A**) Longer (auditory) stimulus durations resulted in lower motor-evoked potential (MEP) amplitudes, regardless of transcranial ultrasonic stimulation (TUS) administration, indicating a sound-duration-dependency of motor inhibitory outcomes (Exp. I). (**B**) A significant effect of auditory stimulus duration was also observed in Experiment III. (**C**) The pitch of auditory stimuli also affected MEPs, where lower amplitudes were observed following a 1 kHz tone compared to a 12 kHz tone. There was no significant effect of TUS. Conventions are as in *Figures 1 and 2C*. *p<0.05, **p<0.01.

sound-sham) and 'stimulus duration' (100/500 ms) revealed only a significant main effect of (auditory) stimulus duration, where longer stimulus durations resulted in stronger MEP attenuation ($F(1,11)$ = 10.07, p=0.009, $\eta_p^2$ = 0.48). There was no significant effect of condition ($F(2,11)$ = 1.30, p=0.311, $\eta_p^2$ = 0.19), nor an interaction between stimulus duration and condition ($F(2,11)$ = 0.65, p=0.543, $\eta_p^2$ = 0.11). These results further demonstrate that the auditory confound and its timing characteristics, rather than ultrasonic neuromodulation, underlies the observed inhibition of motor cortical excitability (*Figure 4A*).

We further tested auditory effects in Experiment III, where we administered sound-sham stimuli at four combinations of duration and pitch. A LMM with factors 'duration' (500/700 ms) and 'pitch' (1/12 kHz) revealed significantly lower MEPs following 500 ms auditory stimuli (*Figure 4*; duration: $F(1,15)$ = 7.12, p=0.017, $\eta_p^2$ = 0.32; pitch: $F(1,15)$ = 0.02, p=0.878, $\eta_p^2$ = 2·10⁻³; interaction: $F(1,15)$ = 2.23, p=0.156, $\eta_p^2$ = 0.13), supporting the role of auditory stimulus timing in perturbation of MEP amplitude.

Subsequently, ultrasonic stimulation was also administered alongside these four auditory stimuli. Here, an LMM with factors 'auditory stimulus duration' (500/700 ms), 'pitch' (1/12 kHz), and 'ultrasonic stimulation' (yes/no) revealed no significant effect of auditory stimulus duration in contrast to the first test ($F(1,15)$ = 0.44, p=0.517, $\eta_p^2$ = 0.03). However, a 1 kHz pitch resulted in significantly lower MEP amplitudes than a 12 kHz pitch (*Figure 4*; $F(1,15)$ = 4.94, p=0.042, $\eta_p^2$ = 0.25). Importantly, we find no evidence for ultrasonic neuromodulation, where both on-target TUS and sound-sham reduced MEP amplitude from baseline (*Figure 1*), and where applying on-target TUS did not significantly affect MEP amplitude as compared to sound-sham ($F(1,15)$ = 0.42, p=0.526, $\eta_p^2$ = 0.03; *Figure 4*). We observed a nonsignificant trend for the interaction between 'ultrasonic stimulation' and 'auditory stimulus duration' ($F(1,15)$ = 4.22, p=0.058, $\eta_p^2$ = 0.22). No trends were observed for the remaining

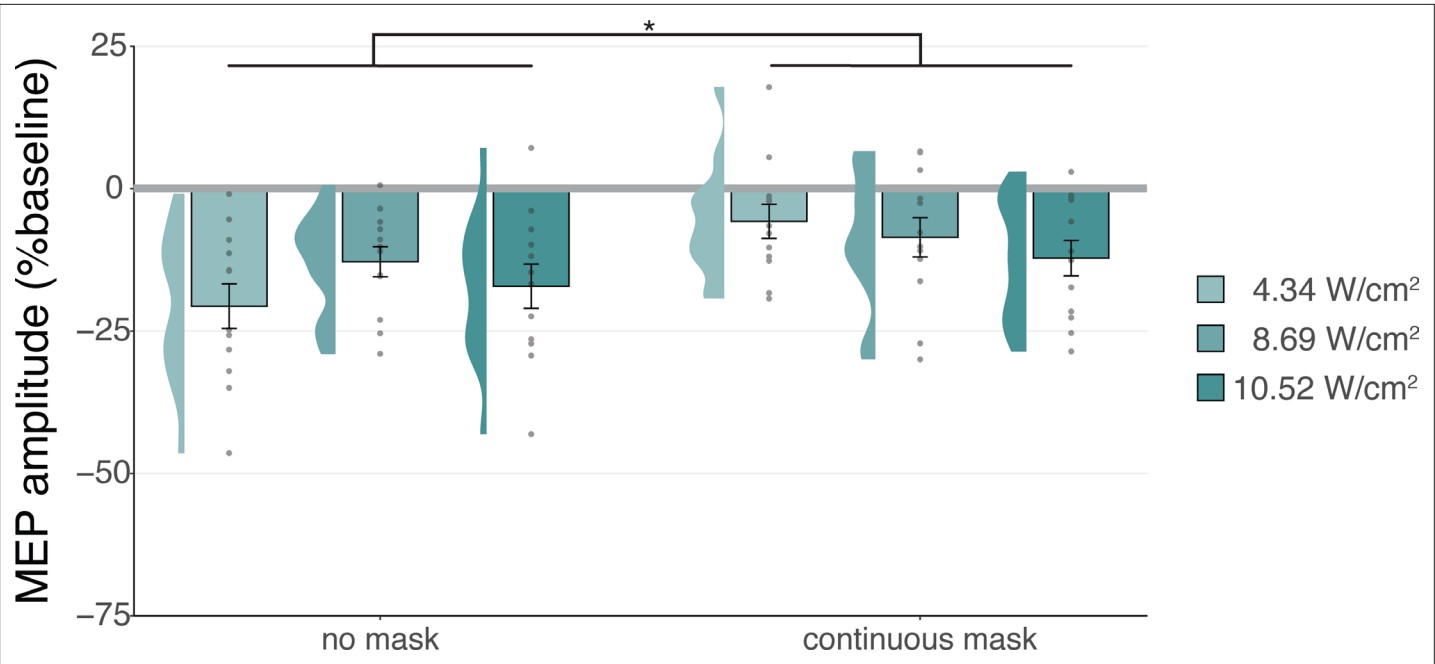

**Figure 5.** Sound-driven effects on corticospinal excitability. Less motor-evoked potentials (MEP) attenuation was measured during continuous masking, particularly for lower stimulation intensities (i.e. auditory confound volumes), pointing towards a role of transcranial ultrasonic stimulation (TUS) audibility in MEP attenuation. *p<0.05.

interactions between these three factors (all $\eta_p^2$ < 0.06, p>0.3). Taken together, these results do not provide evidence for direct ultrasonic neuromodulation but support the influence of auditory stimulation characteristics on motor cortical excitability.

## TUS audibility and confound volume

In Experiment IV, we applied TUS to an inactive target – the white matter ventromedial to the left-hemispheric hand motor area – both with and without a continuous auditory masking stimulus. MEP amplitudes did not significantly differ in baseline conditions regardless of whether a continuous sound was being played (b=0.03, SE = 0.06, t(11) = 0.52, p=0.616), indicating that continuous auditory stimulation alone might not be sufficient to inhibit MEP amplitude.

We additionally applied stimulation at multiple intensities to isolate the effect of auditory confound volume. A linear mixed model with factors 'masking' (no mask/masked) and 'intensity' (4.34/8.69/10.52 Wcm$^{-2}$) with a random intercept and slope for each factor revealed a significant interaction (F(2,4038)=3.43, p=0.033, $\eta_p^2$ = 2·10$^{-3}$) and an accompanying effect of 'masking' with lesser MEP attenuation when stimulation was masked (F(1,11) = 11.84, p=0.005, $\eta_p^2$ = 0.52; *Figure 5*). Follow-up comparisons revealed significantly less attenuation for masked stimulation at 4.34 W/cm$^2$ intensity (F(1,11) = 13.02, p=0.004, $\eta_p^2$ = 0.55), and a nonsignificant trend for the higher intensities (8.69 W/cm$^2$: F(1,11) = 3.87, p=0.077, $\eta_p^2$ = 0.27; 10.52 W/cm$^2$: F(1,11) = 3.47, p=0.089, $\eta_p^2$ = 0.24). In direct comparisons to baseline, all conditions resulted in a significant inhibition of MEP amplitude (all t<–3.36, all p<0.007), with the exception of continuously masked stimulation at the lowest volume, with an intensity of 4.34 W/cm$^2$ I$_{sppa}$ (b=–0.06, SE = 0.03, t(11) = –2.04, p=0.065).

The data indicate that continuous masking reduces motor inhibition, likely by minimizing the audibility of TUS, particularly when applied at a lower stimulation intensity (i.e. auditory confound volume). The remaining motor inhibition observed during masked trials likely owes to, albeit decreased, persistent audibility of TUS during masking. Indeed, MEP attenuation in the masked conditions descriptively scales with participant reports of audibility. This points towards the role of auditory confound volume in motor inhibition (Appendix 4). Nevertheless, one could instead argue that evidence for direct neuromodulation is seen here. This is unlikely for a number of reasons. First, white matter contains a lesser degree of mechanosensitive ion channel expression and there is evidence that neuromodulation of these tracts may occur primarily in the thermal domain (*Guo et al., 2022*;

*Sorum et al., 2021*). Second, Experiment IV lacks sufficient inferential power in the absence of an additional control and must, therefore, be interpreted in tandem with Experiments I-III. These experiments revealed no evidence for direct neuromodulation using equivalent or higher stimulation intensities and directly targeting gray matter while also using multiple control conditions. Therefore, we propose that persistent motor inhibition during masked trials owes to the continued, though reduced, audibility of the confound (Appendix 4). However, future work including an additional control (site) is required to definitively disentangle these alternatives.

### Preparatory cueing of TMS

We find that MEP attenuation results from auditory stimulation rather than direct neuromodulation. Two putative mechanisms through which sound cuing may drive motor inhibition have been proposed, positing either that explicit cueing of TMS timing results in compensatory processes that drive MEP reduction (*Capozio et al., 2021*; *Tran et al., 2021*), or suggesting the evocation of a startle response that leads to global inhibition (*Fisher et al., 2004*; *Furubayashi et al., 2000*; *Ilic et al., 2011*; *Kühn et al., 2004*; *Wessel and Aron, 2013*). Critically, we can dissociate between these theories by exploring the temporal dynamics of MEP attenuation. One would expect a startle response to habituate over time, where MEP attenuation would be reduced during startling initial trials, followed by a normalization throughout the course of the experiment. Alternatively, if temporally contingent sound-cueing of TMS drives inhibition, MEP amplitudes should decrease over time as the relative timing of TUS and TMS is being learned, followed by a stabilization at a decreased MEP amplitude once this relationship has been learned.

In Experiments I and II, linear mixed models with 'trial number' as a predictor show significant changes in MEP amplitude throughout the experiment, pointing to a learning effect. Specifically, in Experiment I, a significant reduction in MEP amplitude was observed across the first 10 trials where a 500 ms stimulus was delivered ($b=-0.04$, SE = 0.01, $t(11) = -2.88$, p=0.015), followed by a stabilization in subsequent blocks ($b = -2 \cdot 10^{-4}$, SE = $3 \cdot 10^{-4}$, $t(11) = -0.54$, p=0.601). This same pattern was observed in Experiment II, with a significant reduction across the first 20 trials ($b=-0.01$, SE = $3 \cdot 10^{-3}$, $t(26) = -4.08$, p=$4 \cdot 10^{-4}$), followed by stabilization ($b=6 \cdot 10^{-5}$, SE = $1 \cdot 10^{-4}$, $t(26) = 0.46$, p=0.650; *Figure 6*). The data suggest that the relative timing of TUS and TMS is learned across initial trials,

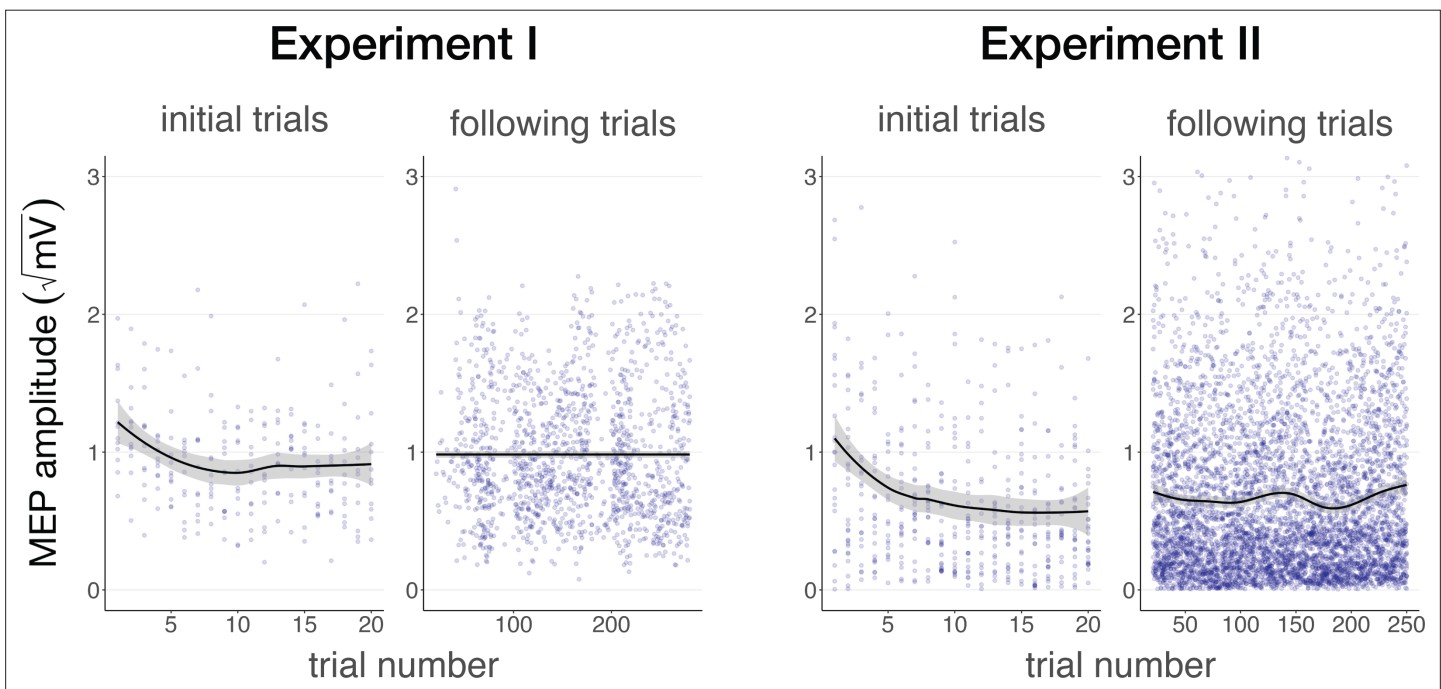

**Figure 6.** Auditory cueing of transcranial magnetic stimulation (TMS). There was a significant reduction in motor-evoked potential (MEP) amplitude when participants were first presented with a 500 ms stimulus (initial trials) in Experiment I (left) and Experiment II (right), followed by a stabilization of MEP amplitude during the rest of the experiment (following trials), indicating a learning process in which TUS acts as a cue that signals the onset of TMS. The solid line depicts the loess regression fit, and the shaded area represents the 95% confidence interval.

followed by a stabilization at a decreased MEP amplitude once this relationship has been learned. These results could reflect auditory cueing of TMS that leads to a compensatory expectation-based reduction of motor excitability.

## Discussion

In this study, we show the considerable impact of auditory confounds during audibly pulsed TUS in humans. We employed improved control conditions compared to prior work across four experiments, one preregistered, at three independent institutions. Here, we disentangle direct neuromodulatory and indirect auditory contributions during ultrasonic neuromodulation of corticospinal excitability. While we corroborated motor inhibitory effects of online TUS (*Fomenko et al., 2020*; *Legon et al., 2018b*; *Xia et al., 2021*), we demonstrated that this inhibition also occurs with stimulation of a control region or presentation of a sound alone, suggesting that the auditory confound rather than direct ultrasonic neuromodulation drives inhibition. Furthermore, no direct neuromodulatory effects on overall excitability were observed, regardless of stimulation timing, intensity, or masking. However, we note that an exploratory investigation of temporal dynamics indicated ultrasound might introduce noise to the neural system. Importantly, we found convincing evidence that characteristics of auditory stimuli do globally affect motor excitability, where auditory cueing of TMS pulse timing can affect measures of corticospinal excitability. This underscores the importance of explicit cueing in TMS experimental design. Most importantly, our results call for a reevaluation of earlier findings following audible TUS, and highlight the importance of suitable controls in experimental design (*Bergmann and Hartwigsen, 2021*; *Siebner et al., 2022*).

### No evidence for direct neuromodulation by TUS

Prior studies have highlighted sound-driven effects of TUS in behavioral and electrophysiological research (*Airan and Butts Pauly, 2018*; *Braun et al., 2020*; *Guo et al., 2018*; *Johnstone et al., 2021*; *Sato et al., 2018*). Here, we assessed whether the auditory confound of a conventional 1000 Hz pulsed protocol might underlie motor inhibitory effects, which are among the most robust and replicable human findings (*Fomenko et al., 2020*; *Legon et al., 2018b*; *Xia et al., 2021*). While we successfully replicated this inhibitory effect, we found the same inhibition following stimulation of a motor control site (contralateral, active) and stimulation of a white-matter control site (ipsilateral, inactive; *Figure 1*). This contrasts with a prior TUS-TMS study which found that TUS of the contralateral hand motor area did not change motor cortical excitability (*Xia et al., 2021*). Indeed, in all direct comparisons between on-target and control stimulation, no differences in excitability were observed, pointing towards a spatially nonspecific effect of TUS. Considering further inhibitory effects following the administration of an auditory stimulus alone, the data suggest that online TUS motor inhibition is largely driven by the salient auditory confound, rather than spatially specific and direct neuromodulation. However, an exploratory analysis that tested for effects beyond a global shift in excitation-inhibition balance revealed that TUS might interact with ongoing neural dynamics by introducing dose-dependent noise (*Figure 2D*).

We found no evidence of a dose-response relationship between TUS intensity ($I_{sppa}$) and motor inhibition when applying stimulation at a wide range of intensities, nor when testing the relationship between simulated intracranial intensities and changes in excitability (*Figure 2A–C*). Similarly, the administration of a time-locked auditory masking stimulus that effectively reduced TUS detection rates did not provide evidence of direct effects being obscured by audible differences between conditions (*Figure 3*, Appendix 3). Taken together, this study presents no evidence for direct and spatially specific TUS inhibition of motor excitability when applying a clearly audible protocol, despite using improved control conditions, higher stimulation intensities, and a larger sample size than prior studies (*Fomenko et al., 2020*; *Legon et al., 2018b*; *Xia et al., 2021*). Building on these results, the current challenge is to develop efficacious neuromodulatory protocols with minimal auditory interference. Efforts in this direction are already underway (*Mohammadjavadi et al., 2019*; *Nakajima et al., 2022*; *Zeng et al., 2022*).

### Sound-cued motor inhibition

Until now, it was unclear how TUS induced motor inhibition in humans. Here, we show that this inhibition is caused by peripheral auditory stimulation. It is well-known that MEPs are sensitive to both

sensory and psychological factors (*Duecker et al., 2013*). For example, several studies find MEP attenuation following a startling auditory stimulus (*Fisher et al., 2004*; *Furubayashi et al., 2000*; *Ilic et al., 2011*; *Kühn et al., 2004*; *Wessel and Aron, 2013*), and have demonstrated the impact of stimulus duration and volume on this inhibition (*Furubayashi et al., 2000*). It is possible that a similar mechanism is at play for audible TUS protocols. Indeed, we observed modulation of motor cortical excitability dependent upon the characteristics of auditory stimuli, including their duration and timing (*Figure 4A and B*), their pitch/frequency (*Figure 4C*), and whether the confound was audible in general, including perceived volume (*Figure 5*, Appendix 4).

One possible interpretation of the observed MEP attenuation is that the auditory confound acts as a salient cue to predict the upcoming TMS pulse. Prediction-based attenuation has been reported in both sensory and motor domains (*Ford et al., 2007*; *Tran et al., 2021*). For example, MEPs are suppressed when the timing of a TMS pulse can be predicted by a warning cue (*Capozio et al., 2021*; *Tran et al., 2021*). In the current experimental setup, participants could also learn the relative timing of the auditory stimulus and the TMS pulse. Indeed, we observe MEP attenuation emerge across initial trials as participants learn when to expect TMS, until a stable (i.e. learned) state is reached (*Figure 6*). Moreover, no motor inhibition was observed when TUS onset was inaudible or when stimulation timing was potentially too fast to function as a predictive cue (100 ms). Taken together, a parsimonious explanation is expectation-based inhibition of TMS-induced MEPs. This inhibitory response might either reflect the inhibition of competing motor programs – a component of motor preparation – or a homeostatic process anticipating the TMS-induced excitation (*Capozio et al., 2021*; *Tran et al., 2021*).

## Limitations

The precise biomolecular and neurophysiological mechanisms underlying ultrasonic neuromodulation remain under steadily progressing investigation (*Weinreb and Moses, 2022*; *Yoo et al., 2022*). A shared interpretation is that mechano-electrophysiological energy transfer is proportional to acoustic radiation force, and thus proportional to stimulation intensity. Accordingly, one could argue that the TUS dose in the present study could have been insufficient to evoke direct neuromodulation. Indeed, despite the applied intensities exceeding prior relevant human work (*Fomenko et al., 2020*; *Legon et al., 2018b*; *Xia et al., 2021*) the total applied neuromodulatory doses are relatively low as compared to, for example, repetitive TUS protocols (rTUS) in animal work (*Folloni et al., 2019*; *Verhagen et al., 2019*) or recent human studies (*Nakajima et al., 2022*).

Alternatively, insufficient neural recruitment could be attributed to stimulation parameters other than intensity. If so, the absence of direct neuromodulation across these experiments might not generalize to parameters outside the tested set. For example, while we replicated and extended prior work targeting the hand motor area at ~30 mm from the scalp (*Fomenko et al., 2020*; *Legon et al., 2018b*; *Xia et al., 2021*), other studies have suggested that the optimal stimulation depth to engage the hand motor area may be more superficial (*Osada et al., 2022*; *Siebner et al., 2022*).

One might further argue that the TMS hotspot provides insufficient anatomical precision to appropriately target the underlying hand muscle representation with TUS. The motor hotspot may not precisely overly the cortical representation of the assessed muscle due to the increased coil-cortex distance introduced by the TUS transducer. This distance, and the larger TMS coils required to evoke consistent MEPs, results in a broad electric field that is substantially larger than the TUS beam width (e.g. 6 mm for 250 kHz; *Fomenko et al., 2020*; *Legon et al., 2018b*). Thus, it is possible that a transducer aligned with the center of the TMS coil may not be adequate. Nevertheless, we note that previous work utilizing a similar targeting approach has effectively induced changes in corticospinal motor excitability (*Zeng et al., 2022*). We also note that our stimulation depth and targeting procedures were comparable to all prior TUS-TMS studies, and that our simulations confirmed targeting (*Figure 2A*, Appendix 5). In summary, our main finding that the auditory confound drove motor inhibition in the present study, and likely had an impact in previous studies, holds true.

## Considerations and future directions

Crucially, our results do not provide evidence that TUS is globally ineffective at inducing neuromodulation. While the present study and prior research highlight the confounding role of indirect auditory stimulation during pulsed TUS, there remains strong evidence for the efficacy of ultrasonic stimulation

in animal work when auditory confounds are accounted for (*Mohammadjavadi et al., 2019*), or in controlled in-vitro systems such as an isolated retina, brain slices, or neuronal cultures in which the auditory confound carries no influence (*Menz et al., 2013*; *Tyler et al., 2018*).

It follows that where an auditory confound could be expected, appropriate control conditions are critical. These controls could involve stimulating a control region, and/or including a matched sound-only sham. In parallel, or perhaps alternatively, the impact of this confound can be mitigated in several ways. First, we recommend that the influence of auditory components be considered in transducer design and selection. Second, masking the auditory confound can help blind participants to experimental conditions. Titrating auditory mask quality per participant to account for intra- and inter-individual differences in subjective perception of the auditory confound would be beneficial. Here, the method chosen for mask delivery must be considered. While bone-conducting headphones align with the bone conduction mechanism of the auditory confound, they might not deliver sound as clearly as in-ear headphones or speakers. Nevertheless, the latter two rely on air-conducted sound. Notably, in-ear headphones could even amplify the perceived volume of the confound by obstructing the ear canal. Importantly, even when using masking stimuli, auditory stimulation could still influence cognitive task performance, among other measures. Alternative approaches could circumvent auditory confounds by testing deaf subjects, or perhaps more practically by ramping the ultrasonic pulse to minimize or even eliminate the auditory confound. This approach still requires validation and will only be relevant for protocols with pulses of sufficient duration. Here, one can expect that the experimental control required to account for auditory confounds might also hold for alternative peripheral effects, such as somatosensory confounds. Longer pulse durations are common in offline rTUS paradigms (*Zeng et al., 2022*), with more opportunity for inaudible pulse shaping and the added benefit of separating the time of stimulation from that of measurement. However, appropriate control conditions remain central to making inferences on interventional specificity.

## Conclusion

Transcranial ultrasonic stimulation is rapidly gaining traction as a promising neuromodulatory technique in humans. For TUS to reach its full potential we must identify robust and effective stimulation protocols. Here, we demonstrate that one of the most reliable findings in the human literature – online motor cortical inhibition during a 1000 Hz pulsed protocol – primarily stems from an auditory confound rather than direct neuromodulation. Instead of driving overall inhibition, we found preliminary evidence that TUS might introduce noise to ongoing neural dynamics. Future research must carefully account for peripheral confounding factors to isolate the direct neuromodulatory effects of TUS, thereby enabling the swift and successful implementation of this technology in both research and clinical settings.

# Materials and methods
## Participants

This multicenter study comprised of four experiments conducted independently across three institutions. Experiment I (n=12, 4 female, $M_{age}$ = 25.9, $SD_{age}$ = 4.6; METC: NL76920.091.21) and Experiment II (n=27, 13 female, $M_{age}$ = 24.1, $SD_{age}$ = 3.7; METC: NL80331.091.22) were conducted at the Donders Institute of the Radboud University (the Netherlands). Experiment III was conducted at the Krembil Research Institute (n=16, 8 female, $M_{age}$ = 31.4, $SD_{age}$ = 7.9; Toronto University Health Network Research Ethics Board: 20–5740, Canada), and Experiment IV at the Neuroimaging Centre of the Johannes Gutenberg University Medical Centre Mainz (n=12, 11 female, $M_{age}$ = 23.0, $SD_{age}$ = 2.7, Landesärztekammer Rheinland-Pfalz: 2021–15808_01, Germany). All participants were healthy, right-handed, without a history of psychiatric or neurological disorders, and provided informed consent. Ethical approval was obtained for each experiment.

## Transcranial ultrasonic and magnetic stimulation

Ultrasonic stimulation was delivered with the NeuroFUS system (manufacturer: Sonic Concepts Inc, Bothell, WA, USA; supplier/support: Brainbox Ltd., Cardiff, UK). A radiofrequency amplifier powered a piezoelectric ultrasound transducer via a matching network. Transducers consisted of a two-element annular array. Further transducer specifications are reported in *Appendix 8—table 1*. Ultrasonic

**Table 1.** Ultrasonic stimulation parameters.

| Exp. | f (kHz) | depth (mm) | PD (ms) | PRF (kHz) | DC | PTD (ms) | $I_{sppa}$ (W/cm²) | P (Mpa) | $MI_{tc}$ |
|---|---|---|---|---|---|---|---|---|---|
| I | 500 | 35 | 0.3 | 1000 | 30% | 100/500 | 32.5/65 | 1.02/1.44 | 0.99/1.40 |
| II | 250 | 28 | 0.3 | 1000 | 30% | 500 | 6.35/19.06 | 0.45/0.78 | 0.65/1.12 |
| III | 500 | 30 | 0.1 | 1000 | 10% | 500 | 9.26 | 0.54 | 0.53 |
| IV | 250 | 50* | 0.3 | 1000 | 30% | 400 | 4.34/8.69/10.52 | 0.37/0.53/0.58 | 0.53/0.76/0.83 |

f = fundamental frequency, depth = TPO focus setting for distance of free-water full-width half-maximum from transducer exit plane, PD = pulse duration, PRF = pulse repetition frequency, DC = duty cycle, PTD = pulse train duration, $I_{sppa}$ = spatial-peak pulse-average intensity in free-water, P = pressure, $MI_{tc}$ = transcranial derated mechanical index. The ramp shape for all experiments was rectangular. For estimated intracranial indices for Experiments I & II see **Appendix 5—figure 5**.

*Note: Experiments I-III targeted the hand motor area. Experiment IV targeted the corticospinal white matter.

stimulation parameters were based on those used in prior TUS-TMS studies (**Table 1**, **Figure 7A**; **Fomenko et al., 2020**; **Legon et al., 2018b**; **Xia et al., 2021**). While ramping the pulses can in principle mitigate the auditory confound (**Johnstone et al., 2021**; **Mohammadjavadi et al., 2019**), doing so for such short pulse durations (≤0.3 ms) is not effective. Therefore, we used a rectangular pulse shape to match prior work.

Single-pulse TMS was delivered with a figure-of-eight coil held at 45° from the midline to induce an approximate posterolateral to anteromedial current. The hand motor hotspot and required TMS intensity were determined using standard procedures as outlined in Appendix 6. To apply TUS and TMS concurrently, the ultrasound transducer was affixed to the center of the TMS coil using a custom-made 3D-printed clamp (**Figure 7B**; Experiments I, II, & IV; Experiment III: see **Fomenko et al., 2020**). TMS was triggered 10ms prior to the offset of TUS (**Figure 7C**). Muscular activity was recorded in the first dorsal interosseous (FDI; Experiments I-III) or in the abductor pollicis brevis (APB; Experiment IV) via electromyography with surface adhesive electrodes using a belly-tendon montage (**Appendix 6— table 1**).

In Experiments I, II, and IV, we used online neuronavigation with individual anatomical scans to support target selection and consistent TMS and TUS placement (Localite Biomedical Visualization Systems GmbH, Sankt Augustin, Germany; MRI specifications: **Appendix 8—table 2**). Furthermore, we recorded the position of TUS in Experiments I and II for post-hoc acoustic and thermal simulations.

## Experiment I

On-target TUS was delivered to the left-hemispheric hand motor area to determine the effect of ultrasonic stimulation on corticospinal excitability. We introduced controls that improve upon the sole use of flip-over sham conditions used in prior work. First, we applied active control TUS to the right-hemispheric face motor area, allowing for the assessment of spatially specific effects while also better mimicking on-target peripheral confounds. In addition, we also included a sound-only sham condition that closely resembled the auditory confound (**Figure 8**). Specifically, we generated a 1000 Hz square wave tone with 0.3 ms long pulses using MATLAB. We then added white noise at a signal-to-noise ratio of 14:1. This stimulus was administered to the participant via bone-conducting headphones (AfterShockz Trekz, TX, USA). Finally, we incorporated a baseline condition consisting solely of TMS.

Ultrasonic stimulation was delivered at two pulse train durations (100/500 ms) and at two intensities (32.5/65 W/cm² $I_{sppa}$) to probe a potential dose-response effect. Additionally, with consideration of potentially audible differences between on-target and active control stimulation sites, we applied these conditions both with and without masking stimuli identical to those used during a sound-only sham. Auditory stimuli used for sound sham and/or masking for each experiment are accessible here: 10.5281/zenodo.8374148. See Appendix 7 for an overview of conditions and experimental timing for each experiment.

Conditions were administered in a single-blind inter-subject counterbalanced blocked design while participants were seated at rest. Ultrasound gel was used to couple both transducers to the participant's scalp (Aquasonic 100, Parker Laboratories, NJ, USA). In total, participants completed 14 blocks of 20 trials each. Each trial lasted 6±1 s. Two baseline measurements were completed, the first

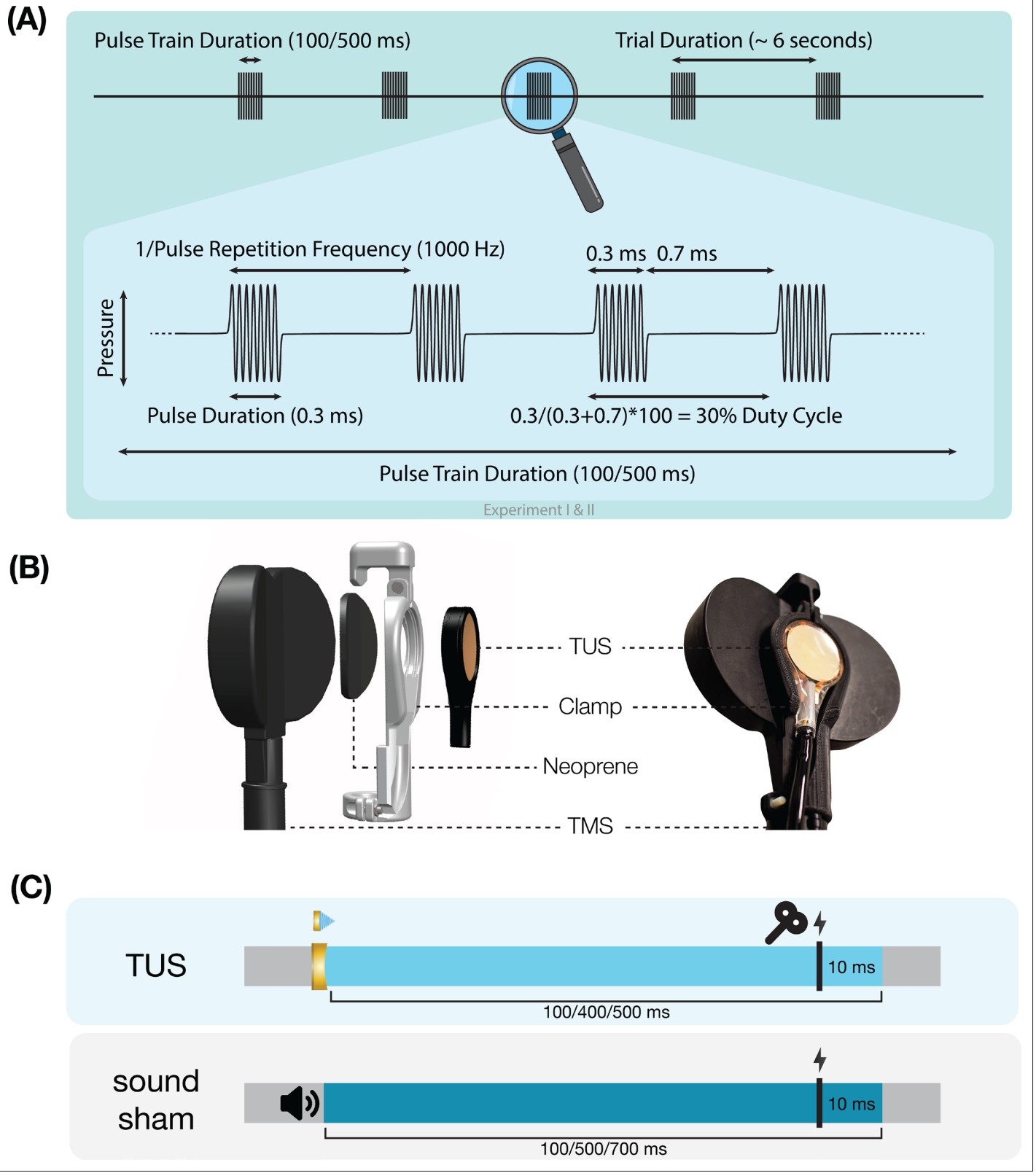

**Figure 7.** Experimental procedures. (**A**) Ultrasonic stimulation protocol for Experiments I & II. In Experiment III a duty cycle of 10% was used. In Experiment IV a stimulus duration of 400 ms was used. (**B**) Transcranial ultrasonic stimulation (TUS)-transcranial magnetic stimulation (TMS) clamp (https://doi.org/10.5281/zenodo.6517599). (**C**) Experimental timing. Detailed experimental timing for each experiment is reported in *Appendix 7—figures 1–4*.

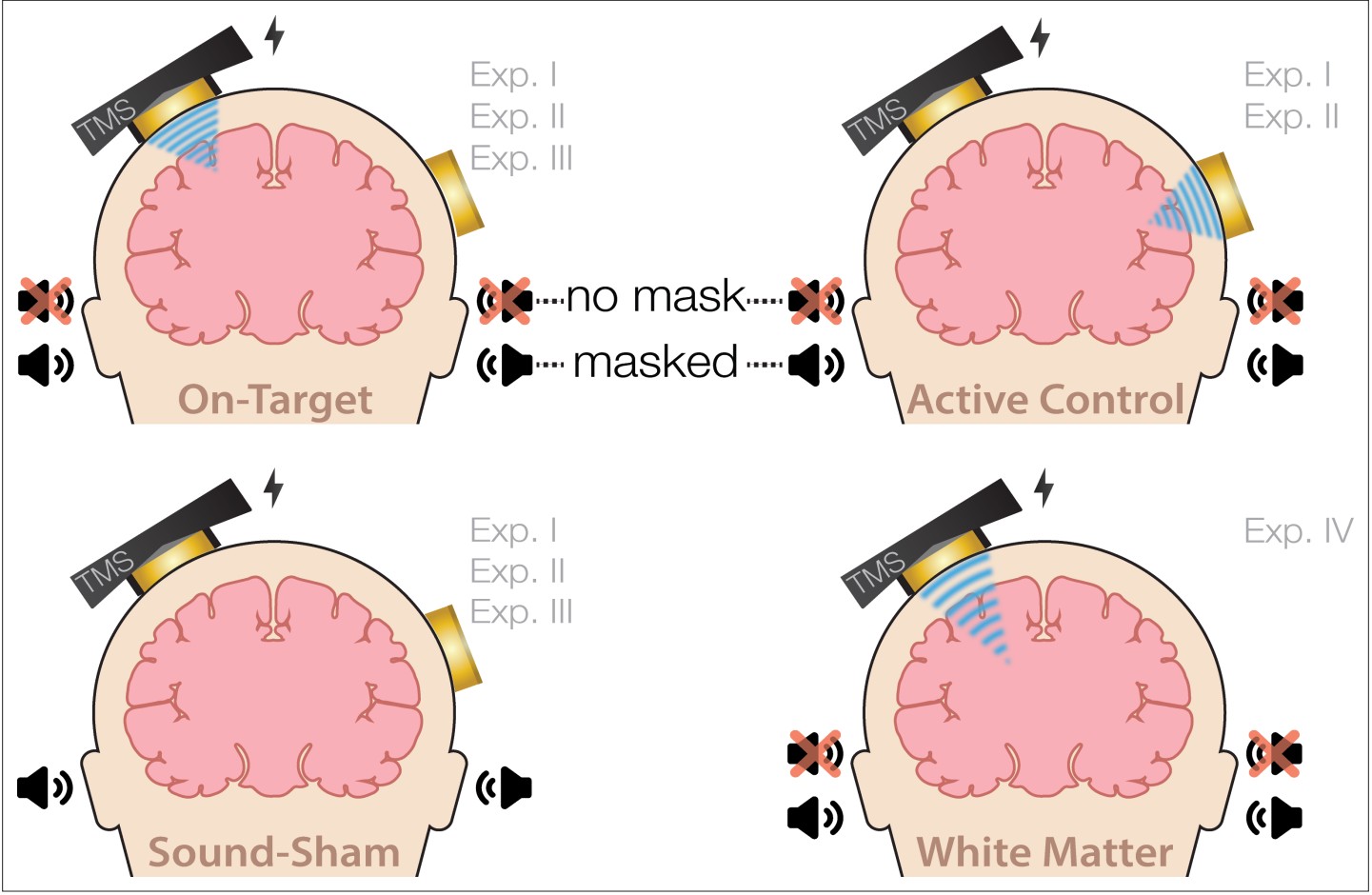

**Figure 8.** Experimental conditions. On-target transcranial ultrasonic stimulation (TUS) of the left-hemispheric hand motor area (Exp. I-III), active control TUS of the right-hemispheric face motor area (Exp. I–II), sound-only sham (Exp. I-III), and inactive control TUS of the white matter ventromedial to the hand motor area (Exp. IV). Conditions involving TUS were presented both with and without auditory masking stimuli.

occurring as one of the first four blocks, and the second as one of the last four, to capture any general shift in excitability throughout the experiment. TMS was administered on every trial for a total of 280 single pulses.

## Experiment II

To confirm and expand upon our findings from Experiment I we conducted a second, preregistered, experiment using the same main conditions and procedures, with a few adaptations (10.17605/OSF. IO/HS8PT). The 2 × 2 × 2 design comprised of stimulation site (on-target/active control), stimulation intensity (6.35/19.06 W/cm²), and auditory masking (no mask/masked). We applied ultrasonic stimulation exclusively at an effective 500 ms pulse train duration. In this experiment, the same 1000 Hz square wave auditory stimulus was used for sound-only sham and auditory masking conditions. This stimulus was administered to the participant over in-ear headphones (ER-3C Insert Earphones, Etymotic Research, Illinois, USA). To better capture any baseline shift in excitability during the experiment, we presented conditions in a single-blind pseudorandomized order in which each consecutive set of 10 trials included each of 10 conditions once. Participants completed 25 trials per condition, resulting in 250 trials total.

To further probe a potential dose-response effect of stimulation intensity, we ran acoustic and thermal simulations (Appendix 5). Here, we assessed the relationship between estimated intracranial intensities and perturbation of corticospinal excitability. While simulations were also run for Experiment I, its sample size was insufficient to test for intracranial dose-response effects.

Following the main experiment, we tested the efficacy of our masking stimuli with a forced-choice task wherein participants reported if they had received TUS for each condition, excluding baseline. Additionally, we investigated whether audible differences between stimulation sites were present during auditory masking (*Appendix 3—figure 1*).

## Experiment III

We further characterized possible effects of auditory confounds on motor cortical excitability by administering varied auditory stimuli, both alongside on-target TUS and without TUS (i.e. sound-only sham). Auditory stimuli were either 500 or 700 ms in duration, the latter beginning 100ms prior to TUS (*Appendix 7—figure 3*). Both durations were presented at two pitches. Using a signal generator (Agilent 33220 A, Keysight Technologies), a 12 kHz sine wave tone was administered over speakers positioned to the left of the participant as in *Fomenko et al., 2020*. Additionally, a 1 kHz square wave tone with 0.5 ms long pulses was administered as in Experiments I, II, IV, and prior research (*Braun et al., 2020*) over noise-cancelling earbuds.

First, we investigated changes in corticospinal excitability from baseline following these auditory stimuli. Participants received 15 trials of baseline (i.e. TMS only) and 15 trials of each of the four sound-only sham stimuli. Conditions were presented in a blocked single-blind randomized order with participants seated at rest. An inter-trial interval of 5 s was used.

Next, we assessed whether applying on-target TUS during these auditory stimuli affected motor excitability. Here, TMS intensity was set to evoke a ~1 mV MEP separately for each of the four sound-only sham conditions (*Appendix 6—figure 2*). To account for different applied TMS intensities between baseline and these conditions, we calculated *Relative MEP amplitude* by multiplying each trial by the ratio of applied TMS intensity to baseline TMS intensity. Participants received 15 trials of each auditory stimulus, once with on-target TUS and once as a sound-only sham. Ultrasound gel (Wavelength MP Blue, Sabel Med, Oldsmar, FL) and a 1.5 mm thick gel pad (Aquaflex, Parker Laboratories, NJ, USA) were used to couple the transducer to the participants' scalps. Conditions were presented in pairs of sound-sham and TUS for each auditory stimulus, counterbalanced between subjects. The order of the different auditory stimuli was randomized across participants.

## Experiment IV

We further investigated the role of TUS audibility on motor excitability by administering stimulation to an inactive control site – the white matter ventromedial to the hand motor area. In doing so, TUS is applied over a homologous region of the scalp and skull without likely direct neuromodulation, thus allowing us to closely replicate the auditory confound while simultaneously isolating its effects.

Here, we probed whether the varying volume of the auditory confound at different stimulation intensities might itself impact motor cortical excitability. To this end, we applied stimulation at 4.34, 8.69, and 10.52 W/cm² $I_{sppa}$, or in effect, at three auditory confound volumes. We additionally applied stimulation both with and without a continuous auditory masking stimulus that sounded similar to the auditory confound. The stimulus consisted of a 1 kHz square wave with 0.3 ms long pulses. This stimulus was presented through wired bone-conducting headphones (LBYSK Wired Bone Conduction Headphones). The volume and signal-to-noise ratio of the masking stimulus were increased until the participant could no longer hear TUS, or until the volume became uncomfortable.

We administered conditions in a single-blind inter-subject randomized block design. Two blocks were measured per condition, each including 30 TUS-TMS trials and an additional 30 TMS-only trials to capture drifts in baseline excitability. These trials were applied in random order within each block with an inter-trial interval of 5±1 s. Ultrasound gel (Aquasonic 100, Parker Laboratories, NJ, USA) and a ~2–3 mm thick gel pad were used to couple the transducer to the participant's scalp (Aquaflex, Parker Laboratories, NJ, USA). During blocks with auditory masking, the mask was played continuously throughout the block. Following each block, participants were asked whether they could hear TUS (yes/no/uncertain).

## Analysis

Raw data were exported to MATLAB, where MEP peak-to-peak amplitude was calculated using a custom script and confirmed by trial-level visual inspection. The data and code are publicly available (https://doi.org/10.34973/jh6z-yh31; *Kop et al., 2024*). Trials where noise prevented an MEP from

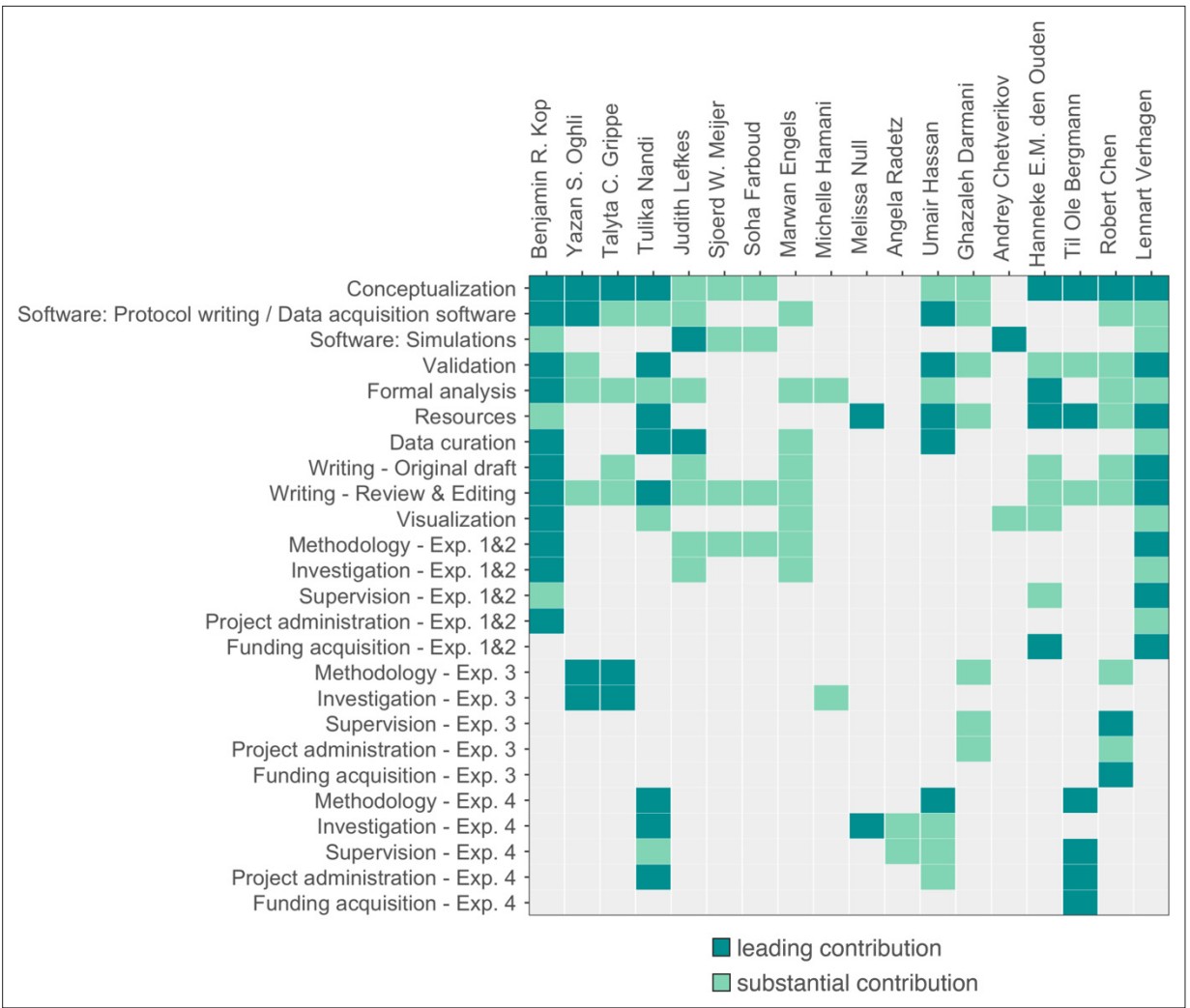

**Figure 9.** Contribution diagram. This figure depicts the involvement of each author using the CRediT taxonomy (**Brand et al., 2015**) and categorizes their contributions according to three levels represented by color: 'none (gray)', 'substantial contribution (light green)', 'leading contribution (dark green)'.

being sufficiently quantified were removed. Given the right-skewed nature of the raw MEP values, we performed a square root transformation to support parametric statistics. For visualization purposes, baseline corrected MEP amplitudes were also calculated.

Linear mixed-effects models (LMMs) were fitted using the lme4 package in R (**Bates et al., 2015**; **R Development Core Team, 2021**). Intercepts and condition differences (slopes) were allowed to vary across participants, including all possible random intercepts, slopes, and correlations in a maximal random effects structure (**Barr et al., 2013**). Statistical significance was set at two-tailed $\alpha$=0.05 and was computed with t-tests using the Satterthwaite approximation of degrees of freedom. For direct comparisons to a reference level (e.g. baseline), we report the intercept ($b$), standard error ($SE$), test-statistics ($t$), and significance ($p$). For main effects and interactions, we report the F statistic, significance, and partial eta squared. LMMs included square root transformed MEP peak-to-peak amplitude as the dependent variable, with the relevant experimental conditions and their interactions as predictors. Given the large number of baseline trials in Experiment IV (50% of the total), the LMM testing effects of stimulation intensity and auditory masking instead included baseline corrected MEP amplitude as the dependent variable.

## CRediT authorship contribution statement

*Figure 9* visualises the contributor roles taxonomy (CRediT) author statement.

## Acknowledgements

Experiments I & II were supported by the Dutch Research Council (NWO), awarding VIDI fellowships to LV (18919) and HEMdO (452-17-016), and the Topsector program Holland High Tech (HiTMaT-38H3). We thank Brittany van Beek for assistance with data acquisition and administration, Sarmad Peymaei for contributions to technical piloting, Norbert Hermesdorf for developing the 3D-printed TUS-TMS clamp, and Gerard van Oijen and Pascal de Water for their technical support. Experiment III was funded by the Canadian Institutes of Health Research (FDN 154292, ENG 173742) and the Natural Science and Engineering Research Council of Canada (RGPIN-2020-04176, RTI-2020-0024). Experiment IV and TN were supported by a grant from the Boehringer Ingelheim Foundation to TOB.

## Additional information

### Competing interests

Umair Hassan: is the head of software development at sync2brain GmbH, where the bossdevice used in Experiment IV was developed. Hanneke EM den Ouden, Lennart Verhagen: Reviewing editor, eLife. The other authors declare that no competing interests exist.

### Funding

| Funder | Grant reference number | Author |
| --- | --- | --- |
| Nederlandse Organisatie voor Wetenschappelijk Onderzoek | 18919 | Lennart Verhagen |
| Nederlandse Organisatie voor Wetenschappelijk Onderzoek | 452-17-016 | Hanneke EM den Ouden |
| Holland High Tech | HiTMaT-38H3 | Lennart Verhagen |
| Canadian Institutes of Health Research | FDN 154292 | Robert Chen |
| Canadian Institutes of Health Research | ENG 173742 | Robert Chen |
| Natural Sciences and Engineering Research Council of Canada | RGPIN-2020-04176 | Robert Chen |
| Natural Sciences and Engineering Research Council of Canada | RTI-2020-0024 | Robert Chen |
| Boehringer Ingelheim Stiftung | | Til Ole Bergmann |

The funders had no role in study design, data collection and interpretation, or the decision to submit the work for publication.

### Author contributions

Benjamin R Kop, Conceptualization, Resources, Data curation, Software, Formal analysis, Supervision, Validation, Investigation, Visualization, Methodology, Writing – original draft, Project administration, Writing – review and editing; Yazan Shamli Oghli, Conceptualization, Software, Formal analysis, Validation, Investigation, Methodology, Writing – review and editing; Talyta C Grippe, Conceptualization, Software, Formal analysis, Investigation, Methodology, Writing – review and editing; Tulika Nandi, Conceptualization, Resources, Data curation, Software, Formal analysis, Supervision, Validation, Investigation, Visualization, Methodology, Project administration, Writing – review and editing; Judith Lefkes, Conceptualization, Data curation, Software, Formal analysis, Investigation, Methodology, Writing – original draft, Writing – review and editing; Sjoerd W Meijer, Soha Farboud, Conceptualization, Software, Methodology, Writing – review and editing; Marwan Engels, Data curation, Software, Formal analysis, Investigation, Visualization, Methodology,

Writing – original draft, Writing – review and editing; Michelle Hamani, Investigation; Melissa Null, Resources, Investigation; Angela Radetz, Supervision, Investigation; Umair Hassan, Conceptualization, Resources, Data curation, Software, Formal analysis, Supervision, Validation, Investigation, Methodology, Project administration; Ghazaleh Darmani, Conceptualization, Resources, Software, Supervision, Validation, Methodology, Project administration; Andrey Chetverikov, Software, Visualization; Hanneke EM den Ouden, Conceptualization, Resources, Formal analysis, Supervision, Funding acquisition, Validation, Visualization, Writing – original draft, Writing – review and editing; Til Ole Bergmann, Conceptualization, Resources, Supervision, Funding acquisition, Validation, Project administration, Writing – review and editing; Robert Chen, Conceptualization, Resources, Software, Formal analysis, Supervision, Funding acquisition, Validation, Methodology, Writing – original draft, Project administration, Writing – review and editing; Lennart Verhagen, Conceptualization, Resources, Data curation, Software, Formal analysis, Supervision, Funding acquisition, Validation, Investigation, Visualization, Methodology, Writing – original draft, Project administration, Writing – review and editing

## Author ORCIDs

Benjamin R Kop ![ORCID] https://orcid.org/0000-0001-7817-5845
Hanneke EM den Ouden ![ORCID] https://orcid.org/0000-0001-7039-5130
Lennart Verhagen ![ORCID] https://orcid.org/0000-0003-3207-7929

## Ethics

All participants provided informed consent. Ethical approval was obtained for each experiment (Experiment I: METC - NL76920.091.21; Experiment II: METC - NL80331.091.22; Experiment III: Toronto University Health Network Research Ethics Board - 20-5740; Experiment IV: Landesärztekammer Rheinland-Pfalz - 2021-15808_01).

Reviewer #1 (Public Review): https://doi.org/10.7554/eLife.88762.3.sa1
Reviewer #2 (Public Review): https://doi.org/10.7554/eLife.88762.3.sa2
Author response https://doi.org/10.7554/eLife.88762.3.sa3

# Additional files

## Supplementary files

MDAR checklist

## Data availability

Data and code are available in the Radboud Data Repository: https://doi.org/10.34973/jh6z-yh31 (CC-BY-4.0). The custom TUS-TMS clamp is open-sourced here: https://doi.org/10.5281/zenodo.6517599. The auditory stimuli used for masking and for sound-sham conditions are open-sourced here: https://doi.org/10.5281/zenodo.8374148.

The following datasets were generated:

| Author(s) | Year | Dataset title | Dataset URL | Database and Identifier |
|---|---|---|---|---|
| Kop B, Shamli Oghli Y, Grippe TC, Nandi T, Lefkes J, Meijer SW, Farboud S, Engels M, Hamani M, Null M, Radetz A, Hassan U, Darmani G, Chetverikov A, Ouden HEM, Bergmann TO, Chen R, Verhagen L | 2024 | Auditory confounds can drive online effects of transcranial ultrasonic stimulation in humans | https://doi.org/10.34973/jh6z-yh31 | Radboud Repository, 10.34973/jh6z-yh31 |

*Continued on next page*

*Continued*

| Author(s) | Year | Dataset title | Dataset URL | Database and Identifier |
|---|---|---|---|---|
| Kop B | 2022 | benjamin-kop/TUS-TMS-Clamp: TUS-TMS-Clamp | https://zenodo.org/records/6517599 | Zenodo, 10.5281/zenodo.6517599 |
| Kop B | 2023 | benjamin-kop/auditory-masking-stimuli: Auditory masking stimuli used in 'Auditory confounds can drive online effects of transcranial ultrasonics stimulation in humans' | https://zenodo.org/records/8374148 | Zenodo, 10.5281/zenodo.8374148 |

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

# Appendix 1

## No evidence for direct intracranial dose-response effects (Exp. II)

In Experiments I & II, we tested the potential dose-response effects of TUS by applying stimulation at multiple free-water intensities. Here, no significant effect of administered intensity was observed. However, the efficacy of TUS likely depends on realized intracranial intensities. Therefore, we ran 3D acoustic simulations to estimate the intracranial intensity during on-target TUS. It was not appropriate to combine data from Experiments I and II given the different fundamental frequencies and stimulation depths applied.

Should direct and spatially specific neuromodulation take place, we would expect to see a dose-response effect where MEP amplitude changes with intracranial intensity during on-target TUS, and not during control conditions. Therefore, the critical test to provide evidence of direct neuromodulation is a comparison of the dose-response relationship between on-target TUS and control conditions. To this end, we ran simple linear models for Experiment II, which had a sufficient sample size (n = 27) to assess inter-individual variability. We found no significant difference in the TUS-MEP relationship between on-target TUS and control conditions (active control free-water 6.35 W/cm$^2$: $b$ = -1.02, $SE$ = 3.66, $t$(24) = -0.28, $p$ = 0.783; active control free-water 19.06 W/cm$^2$: $b$ = 0.04, $SE$ = 1.66, $t$(24) = 0.02, $p$ = 0.983; sound-sham free-water 6.35 W/cm$^2$: $b$ = 0.58, $SE$ = 4.78, $t$(24) = 0.12, $p$ = 0.90; sound-sham free-water 19.06 W/cm$^2$: $b$ = -0.66, $SE$ = 1.92, $t$(24) = -0.35, $p$ = 0.733). We conclude that there is no evidence for a direct neuromodulatory intracranial dose-response relationship. This interpretation is in line with our findings when testing the free-water dose-response effects of TUS.

Notably, given that for all individuals the same ultrasound intensity was applied at the source outside the skull, any intracranial differences in intensity are primarily driven by individual skull characteristics, such as skull thickness. These individual characteristics are expected to co-vary with TMS parameters, such as the TMS intensity (% MSO) required to obtain a 1mV MEP. It is conceivable that such co-varying relationships can drive the overall effects of MEP amplitude and MEP inhibition. Indeed, we observe a significant effect of intracranial intensity on the difference in MEP amplitude between on-target TUS at 6.35 W/cm$^2$ and baseline ($b$ = 19.46, $SE$ = 7.91, $t$(24) = 2.46, $p$ = 0.022), as well as a trend for on-target TUS at 19.06 W/cm$^2$ ($b$ = 5.77, $SE$ = 3.01, $t$(24) = 1.92, $p$ = 0.067). Importantly, this is also observed for the sound-sham condition ($b$ = 19.65, $SE$ = 7.58, $t$(24) = 2.59, $p$ = 0.016), where direct neuromodulation is impossible. These observations emphasize the need for rigorous control conditions to support strong inferences of TUS neuromodulation. In summary, in this experiment, no evidence for a specific and direct neuromodulatory dose-response effect was observed.

## (A) Hypothetical effects

### ultrasonic motor inhibition

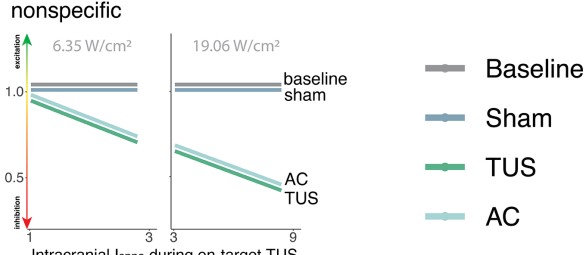

target-specific

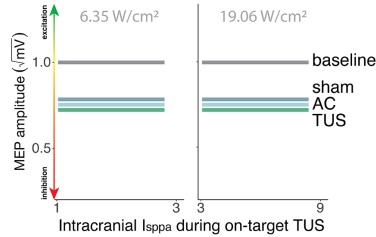

nonspecific

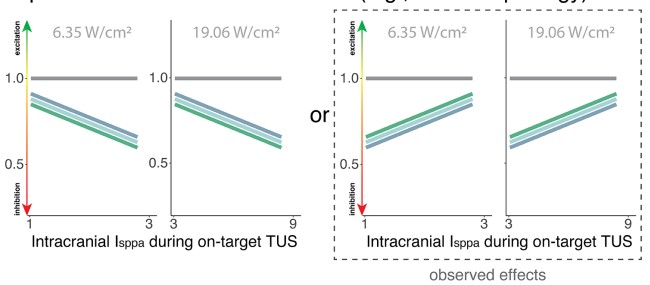

### sound-driven motor inhibition

independent of individual differences

dependent on individual differences (e.g., skull morphology)

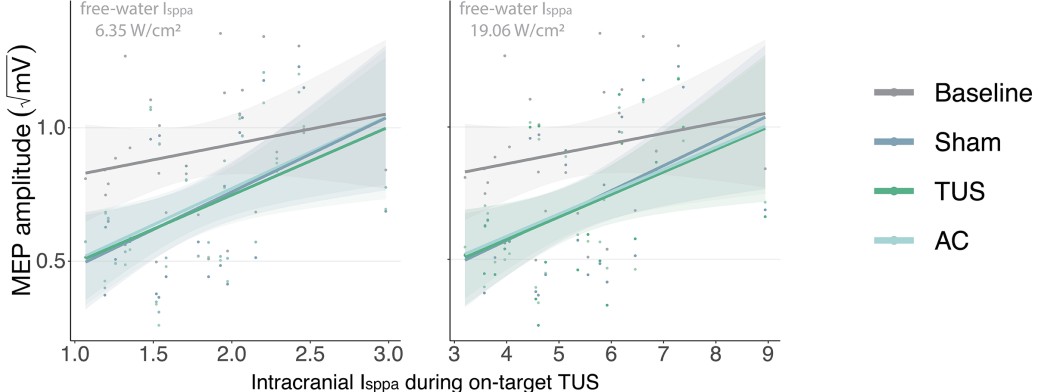

observed effects

## (B) Observed effects

**Appendix 1—figure 1.** No evidence for direct intracranial dose-response effects (Exp. II). (**A**) Hypothetical effects. A target-specific dose-response effect of transcranial ultrasonic stimulation (TUS) would be reflected by a change in motor-evoked potential (MEP) amplitude with increasing intracranial intensity only for on-target TUS (top left), whereas a nonspecific effect of TUS would show the same 'dose-response' for both on-target and active control conditions (top right). If there is sound-driven inhibition, a motor inhibitory effect would be observed for on-target, active control, and sound-only sham (bottom left). If sound-driven inhibition would be subject to individual differences, for example in skull morphology which correlates with intracranial intensity, there would be a correlation between MEP amplitude and intracranial intensity that exists for on-target, active control, and sound-sham conditions (bottom right). The latter corresponds with the observed effects. (**B**) Average square root corrected MEP amplitude, plotted separately for baseline, sham, TUS, and active control conditions, across simulated intracranial intensities. For each participant, we estimated the intracranial ultrasound intensity at the hand M1 target, represented on the x-axis, and the average MEP amplitude for all four conditions. Results for on-target and active control delivered at 6.35 W/cm² free-water $I_{sppa}$ are depicted on the left, and at 19.06 W/cm² free-water $I_{sppa}$ are depicted on the right. Please note that, for reference, the baseline and sham conditions are

*Appendix 1—figure 1 continued*
duplicated across both plots. There is no significant difference between on-target TUS and control conditions across inter-individual intracranial intensities. Points represent the average MEP amplitude per participant per condition. The shaded area represents the 95% CI.

## Appendix 2

### Temporal dynamics (Exp. II)

While studies often focus on quantifying the excitatory or inhibitory effects of TUS, it can be informative to additionally examine how TUS affects neural dynamics in the time domain. Here, we conduct a preliminary exploration of whether autocorrelative measures between a given trial ($t$) and its preceding measure of baseline motor cortical excitability ($t$-1) yield insight into the possible introduction of noise by TUS. To this end, we ran a linear mixed model predicting square root corrected MEP amplitude by 'previous baseline' amplitude, as well as 'stimulation site' (on-target/ active control), 'intensity' (6.35/19.06 W/cm$^2$), and 'masking' (no mask/masked). Previous baseline MEP amplitudes were mean-centered and standardized on a participant level. Random intercepts and slopes were included for 'stimulation site,' 'intensity,' 'previous baseline,' and their interaction, permitting model convergence. This test revealed a significant four-way interaction ($F(1,5222) = 26.10$, $p = 3 \cdot 10^{-7}$, $\eta_p^2 = 5 \cdot 10^{-3}$) and an accompanying main effect of previous baseline MEP amplitude ($F(1,26) = 5.70$, $p = 0.024$, $\eta_p^2 = 0.18$). Follow-up LMMs for unmasked and masked stimulation separately both revealed a significant three-way interaction between 'stimulation site,' 'intensity,' and 'previous baseline' (unmasked: $F(1,176) = 7.32$, $p = 0.007$, $\eta_p^2 = 0.04$; masked: $F(1,30) = 12.10$, $p = 0.002$, $\eta_p^2 = 0.28$). Further follow-up LMMs for each level of masking and intensity were used to test the effects of 'stimulation site', 'previous baseline', and their interaction. These LMMs revealed a significant interaction for all but unmasked 19.06 W/cm$^2$ stimulation (no mask 6.35 W/cm$^2$: $F(1,1293) = 8.36$, $p = 0.004$, $\eta_p^2 = 6 \cdot 10^{-3}$; no mask 19.06 W/cm$^2$: $F(1,1283) = 1.13$, $p = 0.288$, $\eta_p^2 = 9 \cdot 10^{-4}$; masked 6.35 W/cm$^2$: $F(1,1287) = 13.43$, $p = 3 \cdot 10^{-4}$, $\eta_p^2 = 0.01$; masked 19.06 W/cm$^2$: $F(1,1282) = 5.76$, $p = 0.017$, $\eta_p^2 = 4 \cdot 10^{-3}$).

*Appendix 2—figure 1* shows the temporal autocorrelation, as indexed by the linear regression slopes of the relationship between MEP amplitude on the previous baseline trial and the current test trial. Statistical inference is drawn at the current trial (zero-lag, indicated with the filled dots and error bars). For visualization only, the autocorrelation is calculated for a range of time lags, shifting the time series of preceding baseline excitability from 15 trials in the past to 15 trials in the future. This shows, as expected, that autocorrelation and the modulation thereof disappear at larger absolute time lags. The main manuscript shows the difference in autocorrelation between on-target and active control TUS (*Figure 2D*; e.g. masked 6.35 W/cm$^2$ on-target TUS slope – masked 6.35 W/cm$^2$ active control slope). For sound-sham and baseline, the difference score was calculated based on the mean overall active control trials.

In sum, these preliminary exploratory analyses could point towards TUS introducing temporally specific neural noise to ongoing neural dynamics in a dose-dependent manner, rather than simply shifting the overall excitation-inhibition balance. One possible explanation for the discrepancy between trials with and without auditory masking is the difference in auditory confound perception, where without masking the confound's volume differs between intensities, while with masking this difference is minimized. Future studies might consider designing experiments such that temporal dynamics of ultrasonic neuromodulation can be captured more robustly, allowing for quantification of possible state-dependent or nondirectional perturbation effects of stimulation.

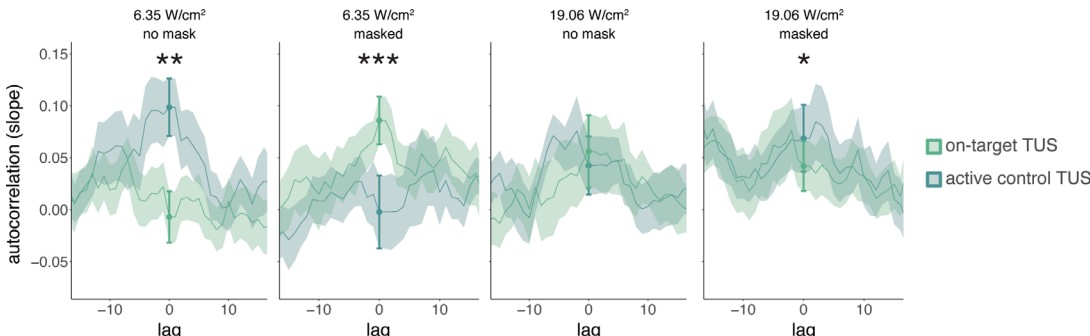

**Appendix 2—figure 1.** Temporal dynamics (Exp. II). We operationalized autocorrelation as the slope between motor-evoked potential (MEP) amplitude on a given trial (t) and its preceding baseline MEP amplitude (t-1) for each condition. Statistical inference is drawn at the current trial (lag = 0). Dots and error bars reflect the mean and

*Appendix 2—figure 1 continued on next page*

*Appendix 2—figure 1 continued*

standard error of the slopes across participants for each level of on-target and active control transcranial ultrasonic stimulation (TUS). Additionally, for visualization purposes, we display autocorrelation for shifts in the preceding baseline time series, with lags ranging from -15 to +15 trials. The line depicts the mean slope across participants for each lag, and the shaded area represents the standard error. *p<0.05, **p<0.01, ***p<0.001.

## Appendix 3

### Blinding efficacy (Exp. II)

In Experiment II, we investigated whether participants were effectively blinded to ultrasonic stimulation when a time-locked auditory masking stimulus was applied (*Appendix 3—figure 1A*). Participants indicated whether they believed TUS was administered across each condition, excluding baseline. Participants completed four trials per condition, resulting in 36 trials in total, presented in pseudorandomized order where each condition is presented randomly in sets of 9 trials. We used a mixed-effects logistic regression to predict participant response (stimulation yes/no) as a function of stimulation site (on-target/active control), free-water intensity (6.35 and 19.05 W/cm²), and masking (no mask/masked) with random intercepts by the participant. This test revealed significantly lower detection rates when masking was applied ($b = -0.90$, $SE = 0.31$, $z = -2.87$, $p = 0.004$), but showed no significant effect of stimulation site ($b = 0.05$, $SE = 0.31$, $z = 0.16$, $p = 0.875$) or intensity ($b = 0.30$, $SE = 0.31$, $z = 0.95$, $p = 0.342$), nor any significant interactions (all $p > 0.6$). Only ultrasonic stimulation applied at higher intensities (19.06 W/cm²) in the absence of an auditory masking stimulus were detected by participants at an above-chance rate (i.e. 50%; on-target: $Z = 2.6$, $p = 0.011$; active control: $Z = 2.9$, $p = 0.004$). Detection rates for lower intensities, masked stimulation, and sound-sham conditions did not differ significantly from chance, as revealed by post-hoc one-sample Wilcoxon signed rank tests (all $Z < 1.7$, $p > 0.09$). Taken together, these results show that the masking used in Experiment II successfully reduced participants' ability to determine whether they had received stimulation.

It is possible that, while blinding to TUS versus no-TUS is successful for explicit judgments, slight audibility differences between on-target and active control stimulation persist even when stimulation is masked. To test for audible differences between stimulation sites (*Appendix 3—figure 1B*), participants were exposed to two consecutive masked TUS trials. These two stimuli were either the same (i.e. [on-target + on-target] or [active control + active control]) or different (i.e. [on-target + active-control/active-control + on-target]). These two trials were presented at both intensities (6.35 and 19.05 W/cm²), to account for the louder auditory confound associated with higher intensity stimulation. Following each pair of trials, participants were asked to rate their similarity on a visual analog scale (0 = very dissimilar, 100 = very similar) across 8 trials in total. We conducted a two-way repeated measures ANOVA predicting similarity rating by stimulation site (same/different) and stimulation intensity (6.35/19.05 W/cm²). Here, masked stimulation over the two different sites sounded less similar than masked stimulation over the same site (*Appendix 3—figure 1B*; $F(1,26) = 14.00$, $p = 9 \cdot 10^{-4}$) while no significant main effect of, or interaction with, intensity was observed (main effect: $F(1,26) = 1.98$, $p = 0.171$; interaction: $F(1,26) = 0.26$, $p = 0.613$). These results suggest that, while masking may effectively blind participants, thus reducing the detection of ultrasonic stimulation, slight audible differences between different stimulation sites may still persist.

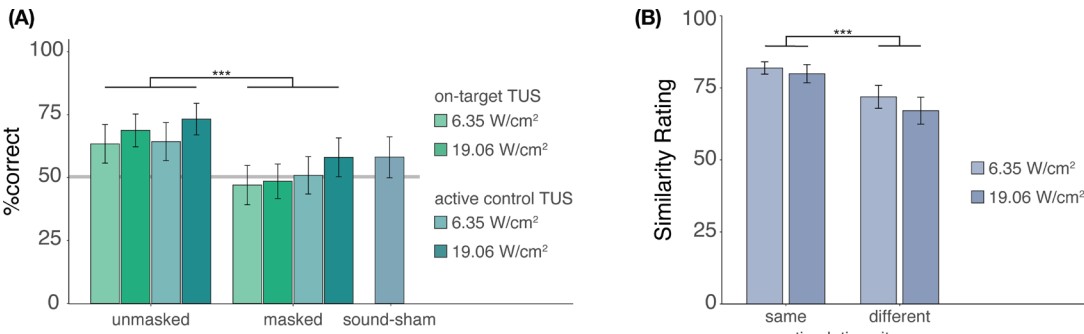

**Appendix 3—figure 1.** Blinding efficacy (Exp. II). (**A**) An auditory masking stimulus significantly decreased participants' ability to determine whether they had received ultrasonic stimulation. (**B**) However, when masked stimulation was applied over different sites, participants still rated these stimuli as sounding less similar than stimulation applied over the same site. Bars represent means across participants, error bars represent the standard error, and points represent individual participants. ***p<0.001.

## Appendix 4

### Subjective report of TUS audibility (Exp. IV)

We propose that the results of Experiment IV reflect the impact of auditory confound volume. Here, the volume of the confound can be regarded as the salience of a cue for the upcoming TMS pulse. When masking is applied, the salience of this cue is diminished. Correspondingly, less motor inhibition was observed when stimulation was masked, as opposed to when no masking stimulus was administered and the auditory confound was clearly audible.

During continuous masking, more motor inhibition was observed at higher auditory confound volumes (i.e. intensities). This is likely because the confound can then be perceived, albeit potentially unconsciously. Indeed, in *Appendix 4—figure 1B*, we also see that participants who rated masked stimulation as uncertain or audible demonstrated more inhibition. Taken together, we suggest that 'dose-response' effects of auditory confound volume are being observed. As the confound approaches and exceeds the boundary of (subconscious) audibility, sufficiently salient cueing of the upcoming TMS pulse takes place, thus evoking proportional motor inhibitory effects.

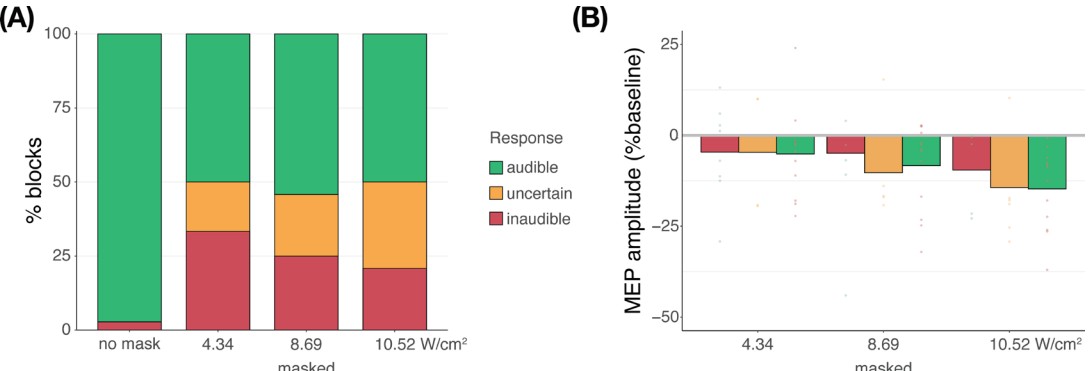

**Appendix 4—figure 1.** Subjective report of transcranial ultrasonic stimulation (TUS) audibility (Exp. IV). (**A**) Depicting the percentage of blocks for which participants reported a condition as audible, uncertain, or inaudible. When no mask was applied, TUS was audible (green) in nearly all cases. During continuous masking, TUS was experienced as inaudible (red) less often at higher stimulation intensities. Descriptively, reports of inaudibility scale with motor-evoked potential (MEP) amplitudes such that in conditions where TUS was inaudible more frequently, less motor inhibition was observed. (**B**) Motor inhibitory effects during masked stimulation, split by audibility. Descriptively, when masking rendered TUS inaudible, less motor inhibition was observed than when participants were uncertain or when stimulation was heard.

# Appendix 5

## Simulations

To obtain estimates of acoustic targeting, realized intracranial dosage, and safety indices, we conducted individualized 3D simulations of acoustic and thermal effects for Experiments I and II using the validated k-Wave MATLAB toolbox, a pseudospectral time-domain solver (*Treeby and Cox, 2010*).

Preprocessing was performed in MATLAB 2019b. Anatomical scans (T1w and T2w) were first segmented into distinct tissues using the SimNIBS headreco tool (*Thielscher et al., 2015*). We then reoriented the segmented volumes such that the axial plane of the recorded transducer location was parallel to the cartesian axis of the computational grid. Subsequently, the segmented volumes were resampled to an isometric volume size of 0.5 mm and interpolated using the nearest neighbor algorithm. We then extracted the skull, scalp, and brain – including gray and white matter – volumes. To correct minor segmentation errors, the volumes were smoothed by a cubic averaging kernel using a window size of four voxels, and re-binarized. The skull layer was additionally combined with a one-voxel outer layer of the filled-in bone volume provided by SimNIBS. Any gaps between the skull and skin volumes were assigned to the skull. The skull, scalp, and brain were all included in acoustic and thermal k-Wave simulations, where all voxels not assigned to these tissues were designated as water. Segmented volumes were then cropped to include the whole skull and the transducer, including a perfectly matched layer (PML) of 10 grid points. The points per wavelength given the 0.5 mm grid spacing were 6 for Experiment I ($f$ = 500 kHz) and 12 for Experiment II ($f$ = 250 kHz).

The acoustic properties of sound speed, density, alpha coefficient, and alpha power, as well as the thermal properties of thermal conductivity and heat capacity, were assigned to the skull, scalp, and brain, as reported in *Appendix 5—table 1* (*Aubry et al., 2022*; *IT'IS Foundation, 2022*).

We modeled on-target TUS with two-element annular arrays using their respective geometric dimensions and fundamental frequencies for Experiments I and II. The position and orientation of the simulated transducer was extrapolated based on the coordinates of the TMS coil recorded during the experiment with neuronavigation software. We calibrated the simulated source amplitude and phase in free water to reach the applied free-water stimulation intensities of 32.5 and 65 W/cm$^2$ (Experiment I) and 6.35 and 19.06 W/cm$^2$ (Experiment II). We calibrated the simulated transducer such that the center of the acoustic profile's full-length half-maximum was at the applied depth of 35.1 mm (Experiment I) or 28 mm (Experiment II) from the exit plane of the transducer. The source amplitude and phase resulting in a simulated acoustic profile with the minimum error in relation to the water-tank calibrated acoustic profile provided by the manufacturer were used for acoustic and thermal simulations including the tissue volumes.

Acoustic simulations were then run for three TUS cycles after reaching the steady-state pressure distribution per applied free-water intensity per participant. The maximum pressure map computed by k-Wave was then used to calculate the realized intracranial pulse-average intensity, which was calculated as: $I_{pa} = \frac{p^2}{2c\rho}$ , where $p$ is acoustic pressure, $c$ is sound speed, and $\rho$ is density. The mechanical index was calculated based on simulated intracranial pressure as: $MI = \frac{p}{\sqrt{f}}$, where $p$ is the peak intracranial pressure, and $f$ is the fundamental frequency. This formula was also used for Experiments III and IV, but with derated free-water pressures.

The simulated pressure map further served as input for the thermal simulations, where we quantified the maximum thermal rise at each point in the computational grid. Here, simulations were run with a 30% DC for a stimulus duration of 500 ms, corresponding to the applied TUS. To compute a conservative safety estimate of thermal rise, stimulation was split into five 100 ms steps, each with a 30 ms on-period and a 70 ms off-period (actual on-period = 0.3 ms per 1 ms).

**Appendix 5—table 1.** Properties for acoustic and thermal simulations (Exp. I-II).

|  | Density (kg/m³) | Sound speed (ms) | Attenuation coefficient | Thermal conductivity (W/m/°C) | Specific heat capacity (J/kg/°C) |
|---|---|---|---|---|---|
| Water | 994 | 1500 | 0.0 | 0.60 | 4178 |
| Scalp | 1090 | 1610 | 0.4 | 0.37 | 3391 |
| Skull | 1850 | 2800 | 8.0 | 0.32 | 1313 |

*Appendix 5—table 1 Continued on next page*

*Appendix 5—table 1 Continued*

| | Density (kg/m³) | Sound speed (ms) | Attenuation coefficient | Thermal conductivity (W/m/°C) | Specific heat capacity (J/kg/°C) |
|---|---|---|---|---|---|
| Brain | 1046 | 1546 | 0.6 | 0.51 | 3630 |

This table describes the acoustic and thermal properties assigned to each component included in simulations.

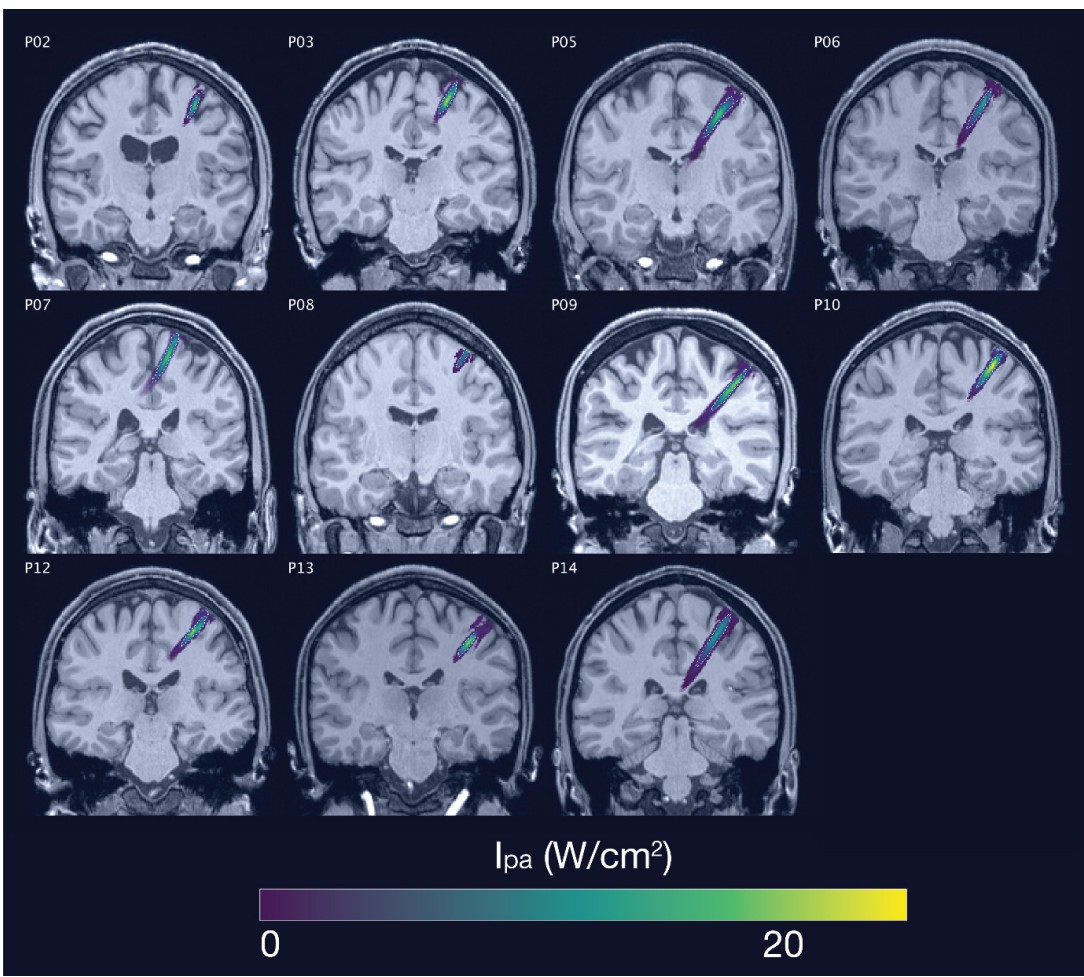

**Appendix 5—figure 1.** Acoustic simulations for Experiment I. Individual simulations for Experiment I of acoustic wave propagation at 65 W/cm² free-water stimulation intensity. Simulated pulse-average intensities above 0.15 W/cm² are depicted. The FWHM of the pressure is indicated by the dashed white line.

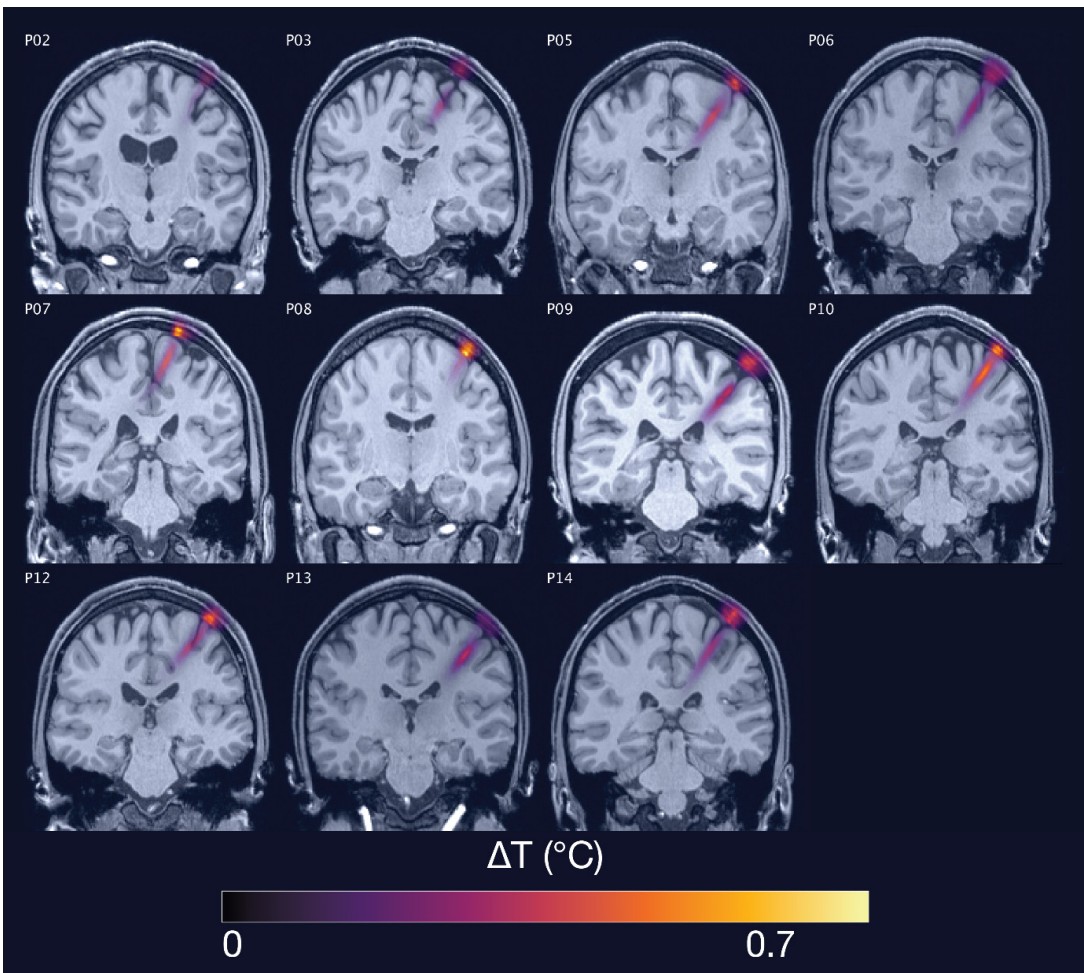

**Appendix 5—figure 2.** Thermal simulations for Experiment I. Individual simulations for Experiment I of thermal rise at 65 W/cm² free-water stimulation intensity.

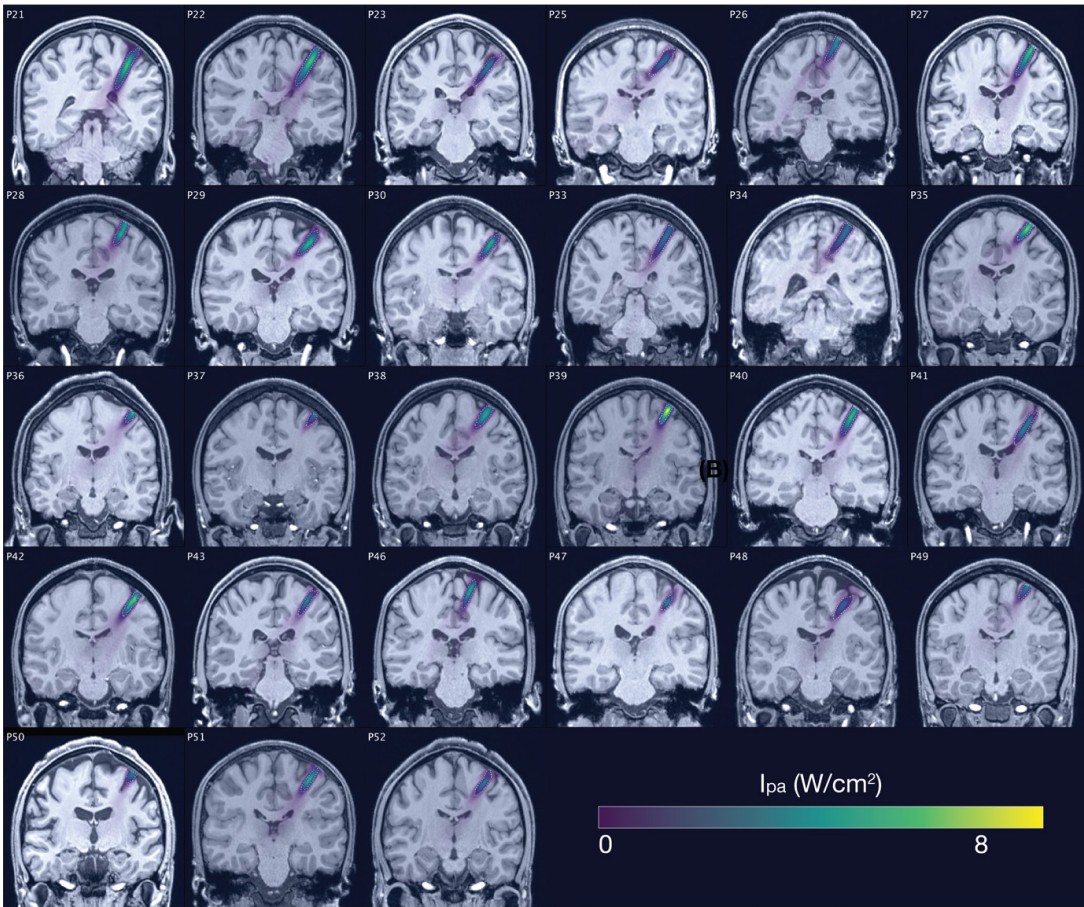

**Appendix 5—figure 3.** Acoustic simulations for Experiment II. Individual simulations for Experiment II of acoustic wave propagation at 19.05 W/cm$^2$ free-water stimulation intensity. Simulated pulse-average intensities above 0.15 W/cm$^2$ are depicted. The FWHM of the pressure is indicated by the dashed white line.

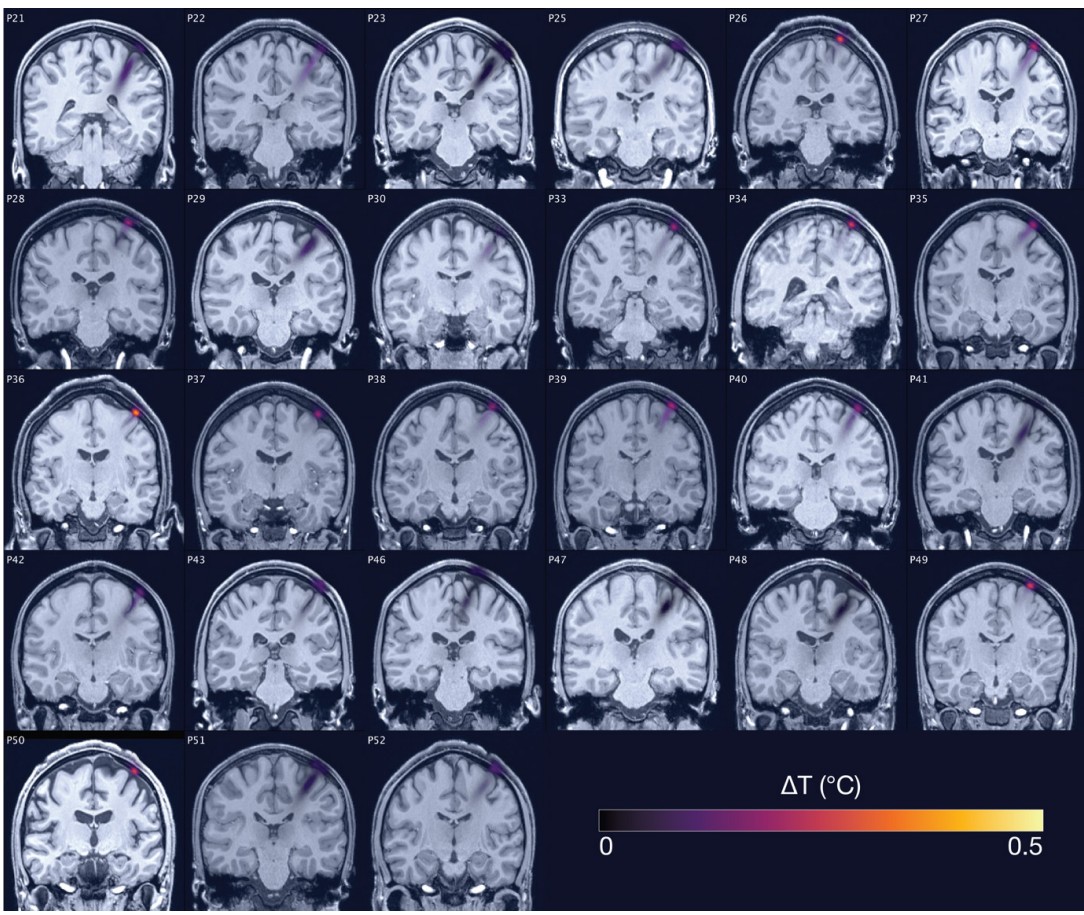

**Appendix 5—figure 4.** Thermal simulations for Experiment II. Individual simulations for Experiment II of thermal rise at 19.05 W/cm² free-water stimulation intensity.

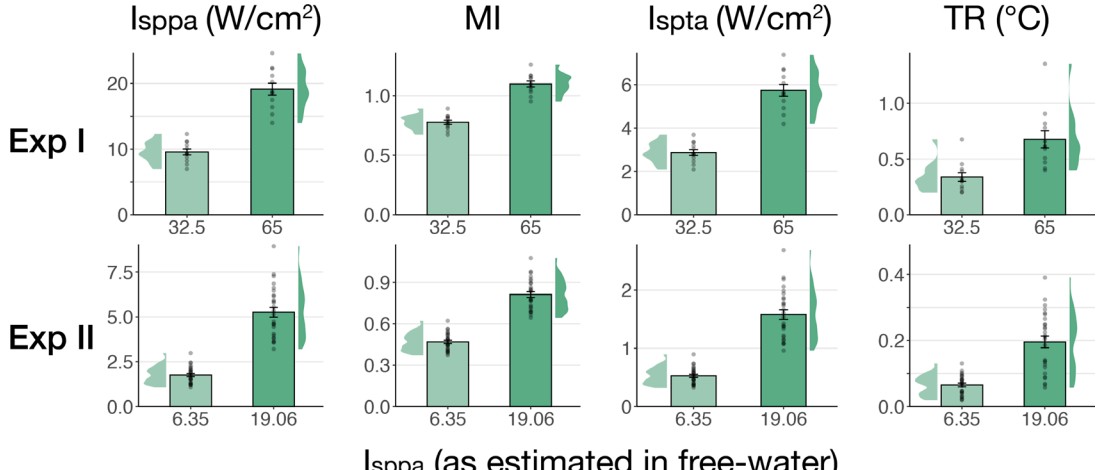

**Appendix 5—figure 5.** Simulated intracranial indices (Exp. I-II). Simulated indices for Experiment I (top) and Experiment II (bottom) for both applied free-water stimulation intensities including spatial-peak pulse- and temporal-average intensity ($I_{sppa}$, $I_{spta}$), the mechanical index (MI), and peak thermal rise (TR).

# Appendix 6

## TMS hotspot and intensity determination

### Experiments I & II

Single-pulse TMS was delivered with a 97 mm figure-of-eight MC-B70 coil powered by a MagPro X100 + MagOption stimulator (MagVenture, Farum, Denmark) using a biphasic pulse shape. The motor hotspot for the first dorsal interosseous (FDI) was determined by positioning the TMS coil over the hand motor area as identified with neuronavigation software, and iteratively adjusting TMS intensity until a consistent MEP was observed of approximately 1 mV amplitude. After probing locations in a ~2 cm radius, the location resulting in the highest and most stable amplitude MEPs was set as the final stimulation site. Next, the percentage of maximum stimulator output (%MSO) was adjusted until an average MEP amplitude of ~1 mV was observed across approximately ten trials. This intensity was used throughout the rest of the experiment. Both TMS and TUS were externally triggered using Signal version 7.05 (CED, Cambridge, UK). The mean percentage of administered TMS maximum stimulator output was 80.4 ± 12.5% for Experiment I, and 88.0 ± 9.3% for Experiment II. These high stimulation intensities are to be expected, given the offset of TMS from the scalp owing to the ultrasound transducer and the TMS current dissipating with distance (*O'Shea and Walsh, 2007*).

### Experiment III

Single-pulse TMS was delivered via a 70 mm figure-of-eight coil powered by a Magstim 200$^2$ stimulator (Magstim, Whitland, Dyfed, UK). The coil was held at 45° from the midline to induce an approximate posterolateral to anteromedial current using a monophasic pulse shape. Optimal positioning was determined by placing the TMS coil over the left-hemispheric hand motor area and moving the coil in increments of 0.5 cm until a clear MEP was observed over the FDI. Following the assessment of optimal coil flatness and orientation, the location and orientation of TMS were marked on the scalp to support consistent placement throughout the experiment. At this site, the minimal intensity required to evoke an average MEP of ~1 mV across ten trials was determined. The mean %MSO was 73.7 ± 11.1% for the baseline condition (i.e. TMS only). The mean %MSO require to evoke a ~ 1mV MEP during the four sound-sham conditions were significantly higher at 77.4 ± 9.4% (1 kHz, 500 ms, $t(15) = -4.64$, $p = 3 \cdot 10^{-4}$), 77.6 ± 10.1% (1 kHz, 700 ms, $t(15) = -4.84$, $p = 2 \cdot 10^{-4}$), 76.9 ± 10.6% (12 kHz, 500 ms, $t(15) = -3.76$, $p = 0.002$), and 77.2 ± 9.8% (12 kHz, 700 ms, $t(15) = -4.09$, $p = 9.7 \cdot 10^{-4}$, see *Appendix 6—figure 2*). To administer stimulation at high temporal precision, both TUS and TMS were externally triggered using Signal 6.04 (CED, Cambridge, UK).

### Experiment IV

The TMS stimulator, coil, pulse shape, and orientation were identical to Experiments I & II, excluding the final four participants in which the Cool-B35 HO coil (Magventure, Farum, Denmark) was used during preparatory measures to allow more precise estimation of the motor hotspot, while always the same TMS coil (MC-B70) was used during the combined TUS-TMS experiment.

The motor hotspot for the abductor pollicis brevis (APB) was determined by positioning the TMS coil over the hand motor area as identified with neuronavigation software, and systematically searching for the largest and most consistent MEP. An automated adaptive staircase procedure with an amplitude threshold of 0.05 mV was used to determine the resting motor threshold (rMT). We subsequently estimated the dose-response curve by applying 20 pulses each at 80, 90, 100, 110, 120, and 130% rMT. In most cases we were unable to reach the plateau of the dose-response curve due to the high motor thresholds resulting from the offset caused by the ultrasound transducer. We aimed to use 120% rMT (n = 3). However, if this intensity surpassed 100% MSO, we opted for 100% MSO instead (n = 9). The mean %MSO was 94.5 ± 10.5%. The TMS intensities required in this experiment were higher than those required in Experiment I-II using the same TMS coil, though still within approximately one standard deviation. This is likely due to the use of a gel pad, which introduces more distance between the TMS coil and the scalp, thus requiring a higher TMS intensity to evoke the same motor activity. To administer stimulation at high temporal precision, both TUS and TMS were externally triggered via a boss-device (sync2brain, Tübingen, Germany) using the BEST Toolbox (*Hassan et al., 2022*).

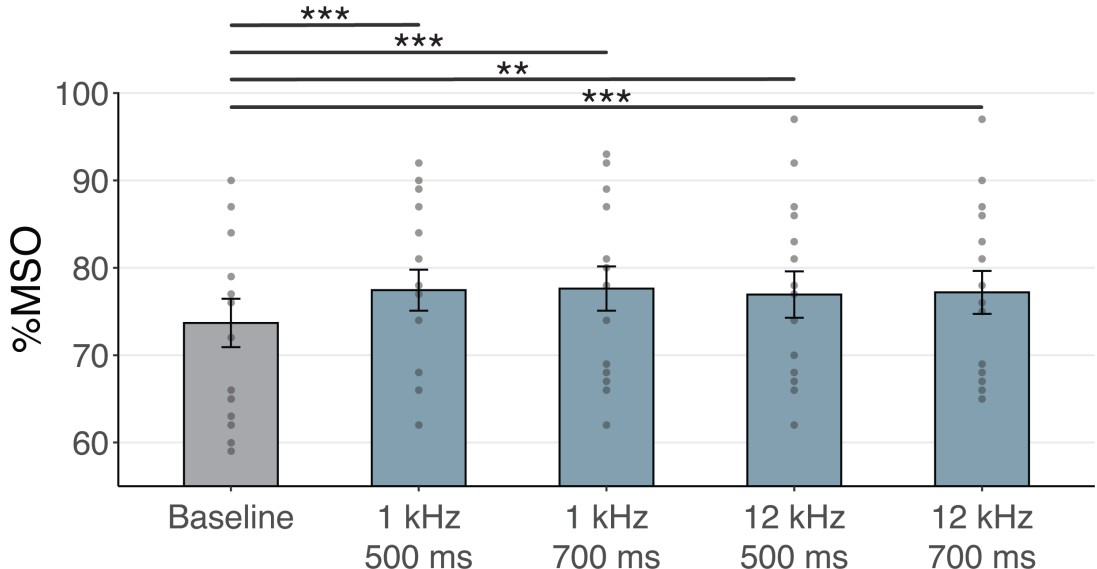

**Appendix 6—figure 1.** Transcranial magnetic stimulation (TMS) hotspot and intensity determination. Percentage of maximum stimulator output (%MSO) is determined by finding the motor hotspot and then adjusting %MSO until a stable 1 mV motor-evoked potential (MEP) is observed (Experiments I-III) or until a MEP of minimally 0.05 mV is observed on at least half of trials (Experiment IV).

**Appendix 6—figure 2.** Transcranial magnetic stimulation (TMS) intensities in Experiment III. Percentage maximum stimulator output (%MSO) required in Experiment III to evoke a 1 mV motor-evoked potential (MEP) at baseline and during the four sound-sham conditions. Each sound-sham condition required a significantly higher %MSO than the baseline to evoke a 1 mV MEP. **p<0.01, ***p<0.001.

**Appendix 6—table 1.** Electromyography.

| Experiment | Muscle | Amplification | Filtering | Digital sampling rate | Software |
|---|---|---|---|---|---|
| I & II | FDI | 1000 gain[*] | 1–1000 Hz[*] | 5 kHz[†] | Signal version 7.05[‡] |
| III | FDI | 1000 gain[§] | 20–2500 Hz[§] | 5 kHz[†] | Signal version 6.04[‡] |
| IV | APB | 10 gain[¶] | <1250 Hz[¶] | 5 kHz[¶] | BEST Toolbox[**] |

This table describes the EMG acquisition parameters used to measure muscular activity in the first dorsal interosseous (FDI) or abductor pollicis brevis (APB) using a belly-tendon montage.

[*]D440-2, Digitimer Ltd., Hertfordshire, UK.

[†]Micro 1401, CED, Cambridge, UK.

[‡]CED, Cambridge, UK.

[§]Intronix Technologies Corporation [model: 2024F], Bolton, Canada.

[¶]Bittium NeurOne Tesla System, Bittium Biosignals Ltd., Finland.

[**]*Hassan et al., 2022*.

## Appendix 7

| | stimulation condition | free-water intensity ($I_{sppa}$) | auditory masking | TUS duration |
|---|---|---|---|---|
| **Exp. I** | on-target TUS | | time-locked to TUS | |
| | active control TUS | 32.5 W/cm² | no mask | 100 ms |
| | sound-sham | 65 W/cm² | masked | 500 ms |
| | baseline | | | |
| **Exp. II** *preregistered* | on-target TUS | | time-locked to TUS | |
| | active control TUS | 6.35 W/cm² | no mask | 500 ms |
| | sound-sham | 19.06 W/cm² | masked | |
| | baseline | | | |
| **Exp. III** | on-target TUS | | time-locked to TUS | |
| | sound-sham | 9.26 W/cm² | masked | 500 ms |
| | baseline | | 1 kHz, 500 ms | |
| | | | 1 kHz, 700 ms | |
| | | | 12 kHz, 500 ms | |
| | | | 12 kHz, 700 ms | |
| **Exp. IV** | inactive control TUS | 4.34 W/cm² | continuous masking | 400 ms |
| | baseline | 8.69 W/cm² | no mask | |
| | | 10.52 W/cm² | masked | |

**Appendix 7—figure 1.** Overview of experimental conditions. For each experiment (row), core experimental conditions are noted. In Experiments I and II, the main conditions included on-target transcranial ultrasonic stimulation (TUS) (left-hemispheric hand motor area), active control TUS (right-hemispheric face motor area), sound-sham (auditory stimulation alone), and baseline (transcranial magnetic stimulation, TMS only). Here, TUS was applied at two stimulation intensities, and both with and without a time-locked auditory masking stimulus. We report the spatial-peak pulse-averaged intensity in free water without attenuation from biological tissue. In Experiment I, TUS was further administered at both 100 and 500 ms stimulus durations. Experiment II is a preregistered study to confirm and extend the findings of Experiment I. In Experiment III, four auditory stimuli of varying pitch and duration were administered both in isolation (sound-sham) and alongside on-target TUS. In Experiment IV, inactive control TUS was administered to the white matter ventromedial to the hand motor area at three intensities (i.e. auditory confound volumes) both with and without a continuous auditory masking stimulus.

# Experiments I & II

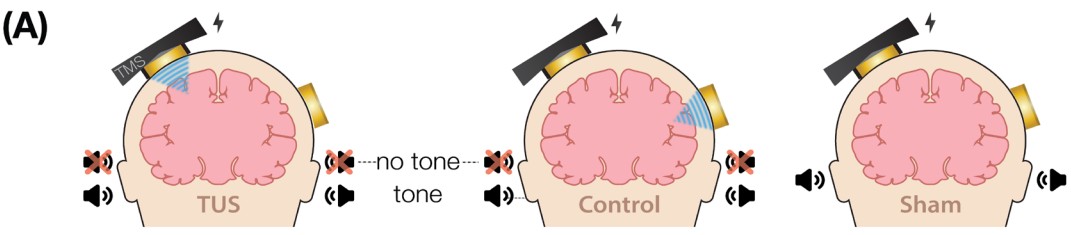

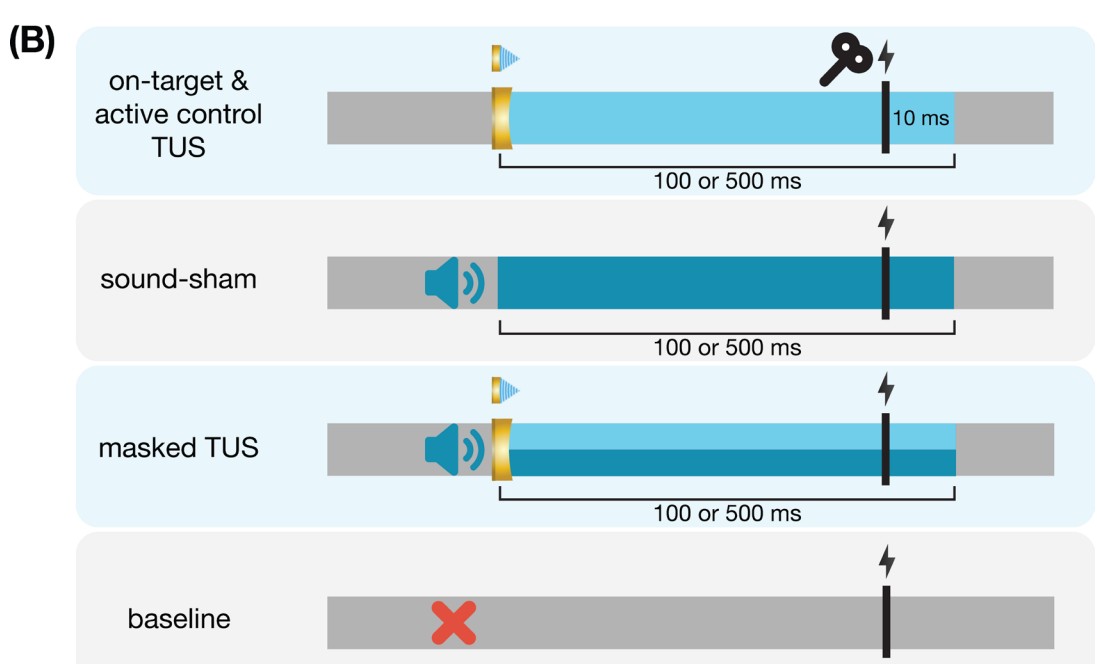

**Appendix 7—figure 2.** Conditions and timing of Experiments I & II. (**A**) On-target left-hemispheric hand area stimulation (left) and active control right-hemispheric face area stimulation (middle) were applied both with and without a masking tone. The sham condition (right) consisted solely of an identical tone. (**B**) On-target and active control transcranial ultrasonic stimulation (TUS) were administered for 100 ms [Experiment I] and 500 ms [Experiments I & II] with transcranial magnetic stimulation (TMS) applied 10 ms prior to TUS-offset. In Experiment I, the auditory stimulus administered during sound-sham and masked conditions began ~50 earlier and was 100 ms longer than TUS, while in Experiment II the auditory stimulus was precisely timed to match TUS. Baseline measurement involved no intervention other than TMS.

# Experiment III

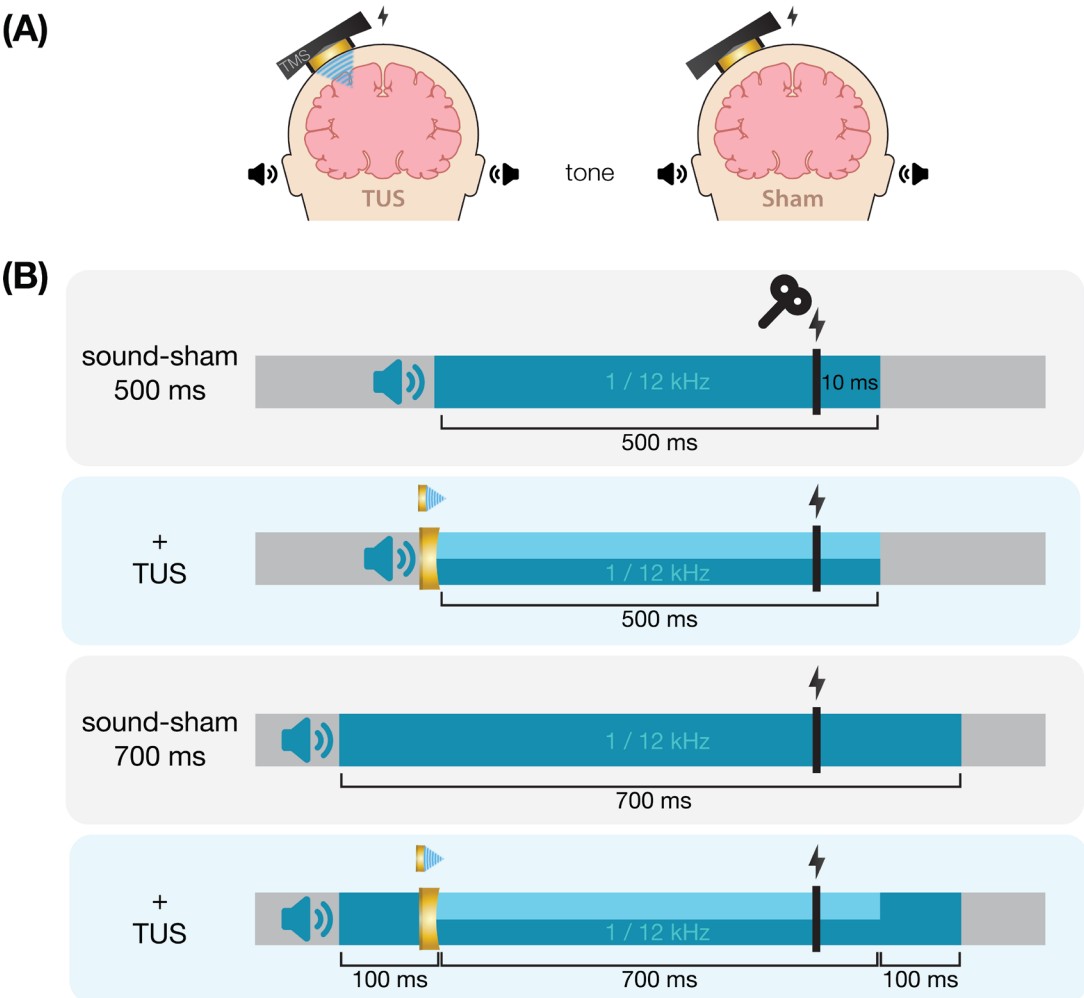

**Appendix 7—figure 3.** Conditions and timing of Experiment III. (**A**) On-target transcranial ultrasonic stimulation (TUS) consisted of left-hemispheric hand motor area stimulation in concurrence with an auditory stimulus (left), while sound-sham consisted solely of an auditory stimulus (right). (**B**) Auditory stimuli were delivered at both 1 and 12 kHz for either 500 or 700 ms. TUS was delivered for 500 ms. Transcranial magnetic stimulation (TMS) was delivered 10 ms prior to TUS offset (i.e. following 490 ms of TUS), and at the same timing in the absence of TUS.

# Experiment IV

**(A)**

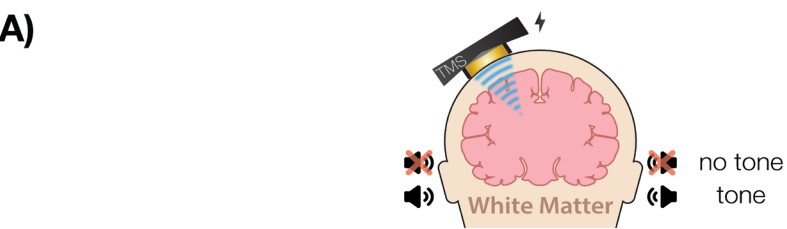

**(B)**

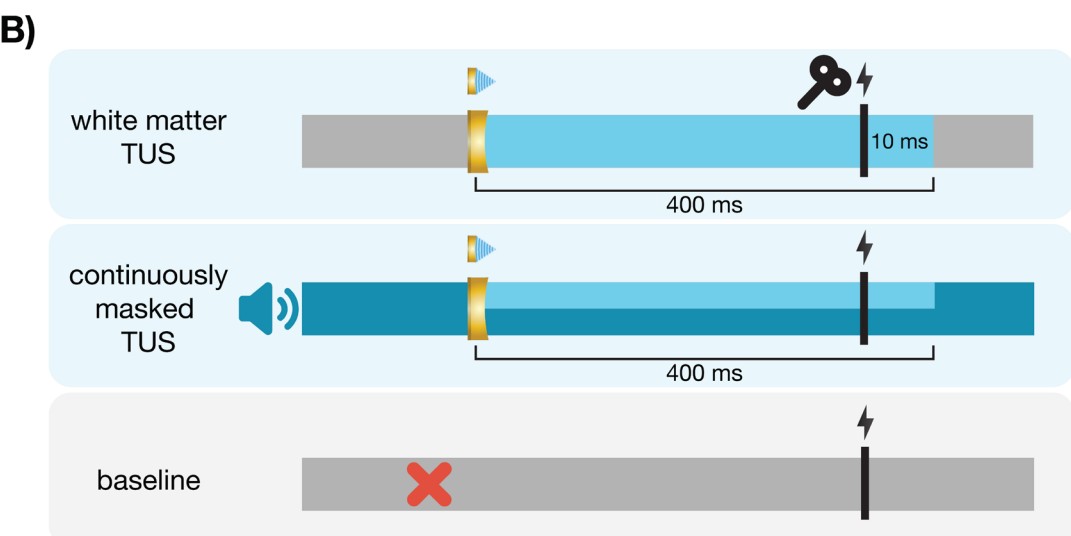

**Appendix 7—figure 4.** Conditions and timing of Experiment III. (**A**) Inactive control stimulation of white matter ventromedial to the left-hemispheric hand motor area. (**B**) White matter transcranial ultrasonic stimulation (TUS) was administered for 400 ms, with transcranial magnetic stimulation (TMS) applied 10 ms prior to TUS offset. In the masked condition, an auditory stimulus was played continuously throughout the entire block.

## Appendix 8

**Appendix 8—table 1.** Transducer specifications.

| Exp. | Model number | N elements | Radius of curvature | Aperature diameter | Width | Solid water coupling |
|---|---|---|---|---|---|---|
| I | CTX500-006 | 2 | 63.2 | 45.0 | ~16 | yes |
| II | CTX250-014 | 2 | 63.2 | 45.0 | ~16 | yes |
| III | H246-01 | 2 | 0 (flat) | 33.6 | 10 | no |
| IV | CTX250-005 | 2 | 63.2 | 45.0 | 16.45 | yes |

This table describes transducer specifications used with a Sonic Concepts TPO (Bothell, WA, USA).

**Appendix 8—table 2.** MRI acquisition parameters.

| Experiment | Scan | TR | TE | FoV read | FoV phase | voxel | N slices |
|---|---|---|---|---|---|---|---|
| I & II | T1w* | 2700 ms | 3.69 ms | 230 mm | 128.1% | 0.9 mm iso | 224 |
| | T2w* | 3200 ms | 408 ms | 230 mm | 128.1% | 0.9 mm iso | 224 |
| IV | T1w† | 2700 ms | 3.7 ms | 260 mm | 88.9% | 0.9 mm iso | 224 |

This table shows the MRI acquisition parameters used in Experiments I, II, and IV. Anatomical T1w scans were used for online neuronavigation. For Experiments I and II, both T1w and T2w scans were used for post-hoc acoustic and thermal simulations.
*3T Siemens Skyra MRI scanner (Siemens Medical Solutions, Erlangen, Germany) with a 32-channel head coil.
†3T Siemens Prisma MRI scanner with a 32-channel head coil.

