## [Editor Report · eLife assessment]

This **important** multicenter study provides **convincing** evidence that the auditory noise emitted during online transcranial ultrasound stimulation (TUS) protocols can pose a considerable confound and is able to explain corticospinal excitability changes as measured with Motor Evoked Potentials (MEP). The findings lay the ground for future studies optimising protocols and control conditions to leverage TUS as a meaningful experimental and clinical tool. A clear strength of the study is the multitude of control conditions (i.e., control sites, acoustic masking, acoustic stimulation). These findings will be of interest to neuroscience researchers using brain stimulation approaches.

---

## [Referee Report · Reviewer #1 (Public Review)]

Summary: The authors have used transcranial magnetic stimulation (TMS) and motor evoked potentials (MEPs) to determine whether the peripheral auditory confound arising from TUS can drive motor inhibition on its own. They gathered data from three international centers in four experiments testing:

- Experiment 1 (n = 11), two different TUS durations and intensities under sound masking or without.- Experiment 2 (n = 27) replicates Exp 1 with different intensities and a fixed TUS duration of 500ms.- Experiment 3 (n = 16) studies the effect of various auditory stimuli testing different duration and pitches while applying TUS in an active site, on-target or no TUS.- Experiment 4 (n = 12) uses an inactive control site to reproduce the sound without effective neuromodulation, while manipulating the volume of the auditory confound at different US intensities with and without continuous sound masking.

Strengths: This study comes from three very strong groups in noninvasive brain stimulation with long experience in neuromodulation, multimodal and electrophysiological recordings. Although complex to understand due to slightly different methodologies across centers, this study provides quantitative evidence relating to the potential auditory confound in online TUS. The results are in line with reductions seen in motor-evoked responses during online 1kHz TUS, and remarkable efforts were made to isolate peripheral confounds from actual neuromodulation factors, highlighting the confounding effect of sound itself.

Weaknesses: However, there are some points that need attention. In my view, the most important are:

1. Despite the main conclusion of the authors stating that there is no dose-response effect of TUS on corticospinal inhibition, the point estimates for change in MEP and Ipssa indicate a more complex picture. The present data and analyses cannot rule out that there is a dose-response function which cannot be fully attributed to difference in sound (since the relationship in inversed, lower intracranial Isppa leads to higher MEP decrease). These results suggest that dose-response function needs to be further studied in future studies.

2. Other methods to test or mask the auditory confound are possible (e.g., smoothed ramped US wave) which could substantially solve part of the sound issue in future studies or experiments in deaf animals etc.

---

## [Referee Report · Reviewer #2 (Public Review)]

Summary:

This study aims to test auditory confounds during transcranial ultrasound stimulation (TUS) protocols that rely on audible frequencies. In several experiments, the authors show that a commonly observed suppression of motor-evoked potentials (MEP) during TUS can be explained by acoustic stimulation. For instance, not only target TUS, but also stimulation of a control site and acoustic stimulation led to suppressed MEP.

The authors have convincingly addressed all of my comments and provided useful additional details. I believe that this is a strong study that will impact the field. Thanks also for making the sound stimuli open-source.

---

## [Author Response]

We are grateful for the insightful suggestions and comments provided by the reviewers. Your constructive feedback has been valuable, and we are thankful for the opportunity to address each point.

We appreciate both reviewers’ recognition of our devotion to rigorous methodology and experimental control in this study, as evidenced by the comments: “remarkable efforts were made to isolate peripheral confounds”, “a clear strength of the study is the multitude of control conditions … that makes results very convincing”, and “thorough design of the study”. Indeed, we hope to have provided more than solid, but compelling evidence for sound-driven motor inhibitory effects of online TUS. We hope that this will be reflected in the assessment. Our conclusions are supported by multiple experiments across multiple institutions using exemplary experimental control including (in)active controls and multiple sound-sham conditions. This contrasts with the sole use of flip-over sham or no-stimulation conditions used in the majority of work to date. Indeed, the current study communicates that substantiated inferences on the efficacy of ultrasonic neuromodulation cannot be made under insufficient experimental control.

In response to the reviewers' comments, we have substantially changed our manuscript. Specifically, we have open-sourced the auditory masking stimuli and specified them in better detail in the text, we have improved the figures to reflect the data more closely, we have clarified the intracranial doseresponse relationship, we have elaborated in the introduction, and we have further discussed the possibility of direct neuromodulation. We hope that you agree these changes have helped to substantially improve the manuscript.

**Public reviews**
(1.1) Despite the main conclusion of the authors stating that there is no dose-response effects of TUS on corticospinal inhibition, both the comparison of Isppa and MEP decrease for Exp 1 and 2, and the linear regression between MEP decrease (relative to baseline) and the estimated Isppa are significant, arguing the opposite, that there is a dose-response function which cannot be fully attributed to difference in sound (since the relationship in inversed, lower intracranial Isppa leads to higher MEP decrease). These results suggest that doseresponse function needs to be further studied in future studies.

We thank the reviewer for bringing up this point. While we are convinced our study provides no evidence for a direct neuromodulatory dose-response relationship, we have realized that the manuscript could benefit from improved clarity on this point.

A dose-response relationship between TUS intensity and motor cortical excitability was assessed by manipulating free-water Isppa (Figure 4C). Here, no significant effect of free-water stimulation intensity was observed for Experiment I or II, thus providing no evidence for a dose-response relationship (Section 3.2). To aid in clarity, ‘N.S.’ has been added to Figure 4C in the revised manuscript.

However, it is likely that the efficacy of TUS would depend on realized intracranial intensity, which we estimated with 3D simulations for on-target stimulation. These simulations resulted in an estimated intracranial intensity for each applied free-water intensity (i.e., 6.35 and 19.06 W/cm2), for each participant. We then tested whether inter-individual differences in intracranial intensity during on-target TUS affected MEP amplitude. We have realized that the original visualization used to display these data and its explanation was unintuitive. Therefore, we have completely revised Supplementary Figure 6. Because of the substantial length of this section, we have not copied it here. Please see the Supplementary material for the implemented improvements.

In brief, we now show MEP amplitudes on the y-axis, rather than expressing values a %change. This plot depicts how individuals with higher intracranial intensities during ontarget TUS exhibit higher MEP amplitudes. However, this same relationship is observed for active control and sound-sham conditions. If there were a direct neuromodulatory doseresponse relationship of TUS, this would be reflected as the difference between on-target and control conditions changing as the estimated intracranial intensity increases. This was not the case. Further, the fact that the difference between on-target stimulation and baseline changes across intracranial intensities is notable, but this occurs to an equal degree in the control conditions. Therefore, these data cannot be interpreted as evidence for a doseresponse relationship.

We hope the changes in Supplementary Figure 6 will make it clear that there is no evidence for direct intracranial dose-response effects.

(1.2) Other methods to test or mask the auditory confound are possible (e.g., smoothed ramped US wave) which could substantially solve part of the sound issue in future studies or experiments in deaf animals etc...

We agree with the reviewer’s statement. We aimed to replicate the findings of online motor cortical inhibition reported in prior work using a 1000 Hz square wave modulation frequency. While ramping can effectively reduce the auditory confound, as noted in the discussion, this is not feasible for the short pulse durations (0.1-0.3 ms) employed in the current study (Johnstone et al., 2021). We have further clarified this point in the methods section of the revised manuscript as follows:

“While ramping the pulses can in principle mitigate the auditory confound (Johnstone et al., 2021; Mohammadjavadi et al., 2019), doing so for such short pulse durations ( <= 0.3 ms) is not effective. Therefore, we used a rectangular pulse shape to match prior work.”

Mitigation of the auditory confound by testing deaf subjects is a valid approach, and has now been added to the revised manuscript in the discussion as follows:

“Alternative approaches could circumvent auditory confounds by testing deaf subjects, or perhaps more practically by ramping the ultrasonic pulse to minimize or even eliminate the auditory confound.”

(1.3) Dose-response function is an extremely important feature for a brain stimulation technique. It was assessed in Exp II by computing the relationship between the estimated intracranial intensities and the modulation of corticospinal excitability (Fig. 3b, 3c). It is not clear why data from Experiment I could not be integrated in a global intracranial dose-response function to explore wider ranges of intracranial intensities and MEP variability.

We chose not to combine data from Experiment 1 in a global intracranial dose-response function because TUS was applied at different fundamental frequencies and focal depths (Experiment I: 500 kHz, 35 mm; Experiment II: 250 kHz, 28 mm). We have now explicitly communicated this under Supplementary Figure 6:

“It was not appropriate to combine data from Experiments I and II given the different fundamental frequencies and stimulation depths applied… we ran simple linear models for Experiment II, which had a sufficient sample size (n = 27) to assess inter-individual variability.”

(1.4) Furthermore, the dose response function as computed with the MEP change relative to baseline shows a significant effect (6.35W/cm2) or a trend (19.06 W/cm2) for a positive linear relationship. This comparison cannot disentangle the auditory confound from the pure neuromodulatory effect but given the direction of the relationship (lower Isppa associated with larger neuromodulatory effect), it is unlikely that it is driven by sound. This relationship is absent for the Active control condition or the Sound Sham condition, more or less matched for peripheral confound. This needs to be further discussed.

Please refer to point 1.1

(1.5) The clear auditory confound arises from TUS pulsing at audible frequencies, which can be highly subject to inter-individual differences. Did the authors individually titrate the auditory mask to account for this intra- and inter-individual variability in auditory perception?

In Experiments I-III, the auditory mask was identical between participants. In Experiment IV, the auditory mask volume and signal-to-noise ratio were adjusted per participant. In the discussion we recommend individualized mask titration. However, we do note that masking successfully blinded participants in Experiment II, despite using uniform masking stimuli (Supplementary Figure 5).

(1.6) How different is the masking quality when using bone-conducting headphones (e.g., Exp. 1) compared to in-ear headphones (e.g., Exp. 2)?

In our experience, bone conducting headphones produce a less clear, fuzzier, sound than in-ear headphones. However, in-ear headphones block the ear canal and likely result in the auditory confound being perceived as louder. We have included this information in the discussion of the revised manuscript:

“Titrating auditory mask quality per participant to account for intra- and inter-individual differences in subjective perception of the auditory confound would be beneficial. Here, the method chosen for mask delivery must be considered. While bone-conducting headphones align with the bone conduction mechanism of the auditory confound, they might not deliver sound as clearly as in-ear headphones or speakers. Nevertheless, the latter two rely on airconducted sound. Notably, in-ear headphones could even amplify the perceived volume of the confound by obstructing the ear canal.”

(1.7) I was not able to find any report on the blinding efficacy of Exp. 1. Do the authors have some data on this?

We do not have blinding data available for Experiment I. Following Experiment I, we decided it would be useful to include such an assessment in Experiment II.

(1.8) Was the possibility to use smoothed ramped US wave form ever tested as a control condition in this set of studies, to eventually reduce audibility? For such fast PRF, for fast PRF, the slope would still need to be steep to stimulate the same power (AUC), it might not be as efficient.

We indeed tested smoothing (ramping) the waveform. There was no perceptible impact on the auditory confound volume. Indeed, prior research has also indicated that ramping over

such short pulse durations is not effective (Johnstone et al., 2021). Taken together, we chose to continue with a square wave modulation as in prior TUS-TMS studies. We have updated the methods section of the manuscript with the following:

“While ramping the pulses can in principle mitigate the auditory confound (Johnstone et al., 2021; Mohammadjavadi et al., 2019), doing so for such short pulse durations ( <= 0.3 ms) is not effective. Therefore, we used a rectangular pulse shape to match prior work.”

Importantly, our research shows that auditory co-stimulation can confound effects on motor excitability, and this likely occurred in multiple seminal TUS studies. While some preliminary work has been done on the efficacy of ramping in humans, future work is needed to determine what ramp shapes and lengths are optimal for reducing the auditory confound.

(1.9) There are other models or experiments that need to be discussed in order to clearly disassociate the TUS effect from the auditory confound effect, for instance, testing deaf animal models or participants, or experiments with multi-region recordings (to rule out the effects of the dense structural connectivity between the auditory cortex and the motor cortex).

The suggestion to consider multi-region recording in future experiments is important. Indeed, the effects of the auditory confound are expected to vary between brain regions. In the primary motor cortex, we observe a learned inhibition, which is perhaps supported by dense structural connectivity with the auditory system. In contrast, in perceptual areas such as the occipital cortex, one might expect tuned attentional effects in response to the auditory cue. We suggest that it is likely that the impact of the auditory confound also operates on a more global network level. It is reasonable to propose that, in a cognitive task for example, the confound will affect task performance and related brain activity, ostensibly regardless of the extent of direct structural connectivity between the auditory cortex and the (stimulated) region of interest.

Regarding the testing of deaf subjects, this has been included in the revised discussion as follows:

“Alternative approaches could circumvent auditory confounds by testing deaf subjects, or perhaps more practically by ramping the ultrasonic pulse to minimize or even eliminate the auditory confound.”

(1.10) The concept of stochastic resonance is interesting but traditionally refers to a mechanism whereby a particular level of noise actually enhances the response of non-linear systems to weak sensory signals. Whether it applies to the motor system when probed with suprathreshold TMS intensities is unclear. Furthermore, whether higher intensities induce higher levels of noise is not straightforward neither considering the massive amount of work coming from other NIBS studies in particular. Noise effects are indeed a function of noise intensity, but exhibit an inverted U-shape dose-response relationship (Potok et al., 2021, eNeuro). In general SR is rather induced with low stimulation intensities in particular in perceptual domain (see Yamasaki et al., 2022, Neuropsychologia). In the same order of ideas, did the authors compare inter-trials variability across the different conditions?

We thank the reviewer for these insightful remarks. Indeed, stochastic resonance is a concept first formalized in the sensory domain. Recently, the same principles have been shown to apply in other domains as well. For example, transcranial electric noise (tRNS) exhibits similar stochastic resonance principles as sensory noise (Van Der Groen & Wenderoth, 2016). Indeed, tRNS has been applied to many cortical targets, including the motor system. In the current manuscript, we raise the question of whether TUS might engage with neuronal activity following principles similar to tRNS. One prediction of this framework would be that TUS might not modulate excitation/inhibition balance overall, but instead exhibit an inverted U-shape dose-dependent relationship with stochastic noise. Please note, we do not use the ‘suprathreshold TMS intensity’ to quantify whether noise could bring a sub-threshold input across the detection threshold, nor whether it could bring a sub-threshold output across the motor threshold. Instead, we use the MEP read-out to estimate the temporally varying excitability itself. We argue that MEP autocorrelation captures the mixture of temporal noise and temporal structure in corticospinal excitability. Building on the non-linear response of neuronal populations, low stochastic noise might strengthen weakly present excitability patterns, while high stochastic noise might override pre-existing excitability. It is therefore not the overall MEP amplitude, but the MEP timeseries that is of interest to us. Here, we observe a non-linear dose-dependent relationship, matching the predicted inverted U-shape. Importantly, we did not intend to assume stochastic resonance principles in the motor domain as a given. We have now clarified in the revised manuscript that we propose a putative framework and regard this as an open question:

“Indeed, human TUS studies have often failed to show a global change in behavioral performance, instead finding TUS effects primarily around the perception threshold where noise might drive stochastic resonance (Butler et al., 2022; Legon et al., 2018). Whether the precise principles of stochastic resonance generalize from the perceptual domain to the current study is an open question, but it is known that neural noise can be introduced by brain stimulation (Van Der Groen & Wenderoth, 2016). It is likely that this noise is statedependent and might not exceed the dynamic range of the intra-subject variability (Silvanto et al., 2007). Therefore, in an exploratory analysis, we exploited the natural structure in corticospinal excitability that exhibits as a strong temporal autocorrelation in MEP amplitude.”

Following the above reasoning, we felt it critical to estimate noise in the timeseries, operationalized as a t-1 autocorrelation, rather than capture inter-trial variability that ignores the timeseries history and requires data aggregation thereby reducing statistical power. Importantly, we would expect the latter index to capture global variability, putatively masking the temporal relationships which we were aiming to test. The reviewer raises an interesting option, inviting us to wonder if inter-trial variability might be sensitive enough, nonetheless. To this end, we compared inter-trial variability as suggested. This was achieved by first calculating the inter-trial variability for each condition, and then running a three-way repeated measures ANOVA on these values with the independent variables matching our autocorrelation analyses, namely, procedure (on-target/active control)*intensity (6.35/19.06)*masking (no mask/masked). This analysis did not reveal any significant interactions or main effects.

**Author response table 1. sa3table1:** 

Effect	DFn	DFd	F	P
intensity	1	26	2.01136951	0.1679998
procedure	1	26	1.51481729	0.2294241
masking	1	26	0.55249790	0.4639630
intensity:procedure	1	26	0.05069027	0.8236267
intensity:masking	1	26	0.17466900	0.6794268
procedure:masking	1	26	0.27552687	0.6040957
intensity:procedure:masking	1	26	1.96443059	0.1728654

(1.11) State-dependency/Autocorrelations: These values were extracted from Exp2 which has baseline trials. Can the authors provide autocorrelation values at baseline, with and without auditory mask? Can the authors comment on the difference between the autocorrelation profiles of the active TUS condition at 6.35W/cm2 or at 19.06W/cm2. They should somehow be similar to my understanding. Besides, the finding that TUS induces noise only when sound is present and at lower intensities is not well discussed.

In the revised manuscript, we have now included baseline in the figure (Figure 4D). Regarding baseline with and without a mask, we must clarify that baseline involves only TMS (no mask), and sham involves TMS + masking stimulus (masked).

The dose-dependent relationship of TUS intensity with autocorrelation is critical. One possible observation would have been that TUS at both intensities decreased autocorrelation, with higher intensities evoking a greater reduction. Here, we would have concluded that TUS introduced noise in a linear fashion.

However, we observed that lower-intensity TUS in fact strengthened pre-existing temporal patterns in excitability (higher autocorrelation), while during higher-intensity TUS these patterns were overridden (lower autocorrelation). This non-linear relationship is not unexpected, given the non-linear responses of neurons.

If this non-linear dependency is driven by TUS, one could expect it to be present during conditions both with and without auditory masking. However, the preparatory inhibition effect of TUS likely depends on the salience of the cue, that is, the auditory confound. In trials without auditory masking, the salience of the confound in highly dependent on (transmitted) intensity, with higher intensities being perceived as louder. In contrast, when trials are masked, the difference in cue salience between lower and higher intensity stimulation in minimized. Therefore, we would expect for any nuanced dose-dependent direct TUS effect to be best detectable when the difference in dose-dependent auditory confound perception is minimized via masking. Indeed, the dose-dependent effect of TUS on autocorrelation is most prominent when the auditory confound is masked.

“In sum, these preliminary exploratory analyses could point towards TUS introducing temporally specific neural noise to ongoing neural dynamics in a dose-dependent manner, rather than simply shifting the overall excitation-inhibition balance. One possible explanation for the discrepancy between trials with and without auditory masking is the difference in auditory confound perception, where without masking the confound’s volume differs between intensities, while with masking this difference is minimized. Future studies might consider designing experiments such that temporal dynamics of ultrasonic neuromodulation can be captured more robustly, allowing for quantification of possible state-dependent or nondirectional perturbation effects of stimulation.”

(1.12) Statistical considerations. Data from Figure 2 are considered in two-by-two comparisons. Why not reporting the ANOVA results testing the main effect of TUS/Auditory conditions as done for Figure 3. Statistical tables of the LMM should be reported.

Full-factorial analyses and main effects for TUS/Auditory conditions are discussed from Section 3.2 onwards. These are the same data supporting Figure 2 (now Figure 3). We would like to note that the main purpose of Figure 2 is to demonstrate to the reader that motor inhibition was observed, thus providing evidence that we replicated motor inhibitory effects of prior studies. A secondary purpose is to visually represent the absence of direct and spatially specific neuromodulation. However, the appropriate analyses to demonstrate this are reported in following sections, from Section 3.2 onwards, and we are concerned that mentioning these analyses earlier will negatively impact comprehensibility.

Statistical tables of the LMMs are provided within the open-sourced data and code reported at the end of the paper, embedded within the output which is accessible as a pdf (i.e., analysis/analysis.pdf).

(1.13) Startle effects: The authors dissociate two mechanisms through which sound cuing can drive motor inhibition, namely some compensatory expectation-based processes or the evocation of a startle response. I find the dissociation somehow artificial. Indeed, it is known that the amplitude of the acoustic startle response habituates to repetitive stimulation. Therefore, sensitization can well explain the stabilization of the MEP amplitude observed after a few trials.

Thank you for bringing this to our attention. Indeed, an acoustic startle response would habituate over repetitive stimulation. A startle response would result in MEP amplitude being significantly altered in early trials. As the participant would habituate to the stimulus, the startle response would decrease. MEP amplitude would then return to baseline levels. However, this is not the pattern we observe. An alternative possibility is that participants learn the temporal contingency between the stimulus and TMS. Here, compensatory expectation-based change in MEP amplitude would be observed. In this scenario, there would be no change in MEP amplitude during early trials because the stimulus has not yet become informative of the TMS pulse timing. However, as participants learn how to predict TMS timing by the stimulus, MEP amplitude would decrease. This is also the pattern we observe in our data. We have clarified these alternatives in the revised manuscript as follows:

“Two putative mechanisms through which sound cuing may drive motor inhibition have been proposed, positing either that explicit cueing of TMS timing results in compensatory processes that drive MEP reduction (Capozio et al., 2021; Tran et al., 2021), or suggesting the evocation of a startle response that leads to global inhibition (Fisher et al., 2004; Furubayashi et al., 2000; Ilic et al., 2011; Kohn et al., 2004; Wessel & Aron, 2013). Critically, we can dissociate between these theories by exploring the temporal dynamics of MEP attenuation. One would expect a startle response to habituate over time, where MEP amplitude would be reduced during startling initial trials, followed by a normalization back to baseline throughout the course of the experiment as participants habituate to the starling stimulus. Alternatively, if temporally contingent sound-cueing of TMS drives inhibition, MEP amplitudes should decrease over time as the relative timing of TUS and TMS is being learned, followed by a stabilization at a decreased MEP amplitude once this relationship has been learned.”

(1.14) Can the authors further motivate the drastic change in intensities between Exp1 and 2? Is it due to the 250-500 carrier difference? It this coming from the loss power at 500kHz?

The change in intensities between Experiments I and II was not an intentional experimental manipulation. Following completion of data acquisition, our TUS system received a firmware update that differentially corrected the 250 kHz and 500 kHz stimulation intensities. In this manuscript, we report the actual free-water intensities applied during our experiments.

(1.15) Exp 3: Did 4 separate blocks of TUS-TMS and normalized for different TMS intensities used with respect to baseline. But how different was it. Why adjusting and then re adjusting intensities?

The TMS intensities required to evoke a 1 mV MEP under the four sound-sham conditions significantly differed from the intensities required for baseline. In the revised appendix, we have now included a figure depicting the TMS intensities for these conditions, as well as statistical tests demonstrating each condition required a significantly higher TMS intensity than baseline.

TMS intensities were re-adjusted to avoid floor effects when assessing the efficacy of ontarget TUS. Sound-sham conditions themselves attenuate MEP amplitude. This is also evident from the higher TMS intensities required to evoke a 1 mV MEP under these conditions. If direct neuromodulation by TUS would have further decreased MEP amplitude, the concern was that effects might not be detectible within such a small range of MEP amplitudes.

(1.16) In Exp 4, TUS targeted the ventromedial WM tract. Since direct electrical stimulation on white matter pathways within the frontal lobe can modulate motor output probably through dense communication along specific white matter pathways (e.g., Vigano et al., 2022, Brain), how did the authors ensure that this condition is really ineffective? Furthermore, the stimulation might have covered a lot more than just white matter. Acoustic and thermal simulations would be helpful here as well.

Thank you for pointing out this possibility. Ultrasonic and electrical stimulation have quite distinct mechanisms of action. Therefore, it is challenging to directly compare these two approaches. There is a small amount of evidence that ultrasonic neuromodulation of white matter tracts is possible. However, the efficacy of white matter modulation is likely much lower, given the substantially lesser degree of mechanosensitive ion channel expression in white matter as opposed to gray matter (Sorum et al., 2020, PNAS). Further, recent work has indicated that ultrasonic neuromodulation of myelinated axonal bundles occurs within the thermal domain (Guo et al., 2022, SciRep), which is not possible with the intensities administered in the current study. Nevertheless, based on Experiment IV in isolation, it cannot be definitively excluded that there TUS induced direct neuromodulatory effects in addition to confounding auditory effects. However, Experiment IV does not possess sufficient inferential power on its own and must be interpreted in tandem with Experiments I-III. Taken together with those findings, it is unlikely that a veridical neuromodulation effect is seen here, given the equivalent or lower stimulation intensities, the substantially deeper stimulation site, and the absence of an additional control condition in Experiment IV. This likelihood is further decreased by the fact that inhibitory effects under masking descriptively scale with the audibility of TUS.

Off-target effects such as unintended co-stimulation of gray matter when targeting white matter is always an important factor to consider. Unfortunately, individualized simulations for Experiment IV are not available. However, the same type of transducer and fundamental frequency was used as in Experiment II, for which we do have simulations. Given the size of the focus and the very low in-situ intensities extending beyond the main focal point, it is incredibly unlikely that effective stimulation was administered outside white matter in a meaningful number of participants. Nevertheless, the reviewer is correct that this can only be directly confirmed with simulations, which remain infeasible due to both technical and practical constraints. We have included the following in the revised manuscript:

“The remaining motor inhibition observed during masked trials likely owes to, albeit decreased, persistent audibility of TUS during masking. Indeed, MEP attenuation in the masked conditions descriptively scale with participant reports of audibility. This points towards a role of auditory confound volume in motor inhibition (Supplementary Fig. 8). Nevertheless, one could instead argue that evidence for direct neuromodulation is seen here. This unlikely for a number of reasons. First, white matter contains a lesser degree of mechanosensitive ion channel expression and there is evidence that neuromodulation of these tracts may occur primarily in the thermal domain (Guo et al., 2022; Sorum et al., 2021). Second, Experiment IV lacks sufficient inferential power in the absence of an additional control and must therefore be interpreted in tandem with Experiments I-III. These experiments revealed no evidence for direct neuromodulation using equivalent or higher stimulation intensities and directly targeting grey matter while also using multiple control conditions. Therefore, we propose that persistent motor inhibition during masked trials owes to continued, though reduced, audibility of the confound (Supplementary Fig. 8). However, future work including an additional control (site) is required to definitively disentangle these alternatives.”

(1.17) Still for Exp 4. the rational for the 100% MSO or 120% or rMT is not clear, especially with respect to Exp 1 and 2. Equipment is similar as well as raw MEPs amplitudes, therefore the different EMG gain might have artificially increased TMS intensities. Could it have impacted the measured neuromodulatory effects?

Experiment IV was conducted independently at a different institute than Experiments I-II. In contrast to Experiments I-II, a gel pad was used to couple TUS to the participant’s head. The increased TMS-to-cortex distance introduced by the gel pad necessitates higher TMS intensities to compensate for the increased offset. In fact, in 9/12 participants, the intended intensity at 120% rMT exceeded the maximum stimulator output. In those cases, we defaulted to the maximum stimulator output (i.e., 100% MSO). We have clarified in the revised supplementary material as follows:

“We aimed to use 120% rMT (n = 3). However, if this intensity surpassed 100% MSO, we opted for 100% MSO instead (n = 9). The mean %MSO was 94.5 ± 10.5%. The TMS intensities required in this experiment were higher than those required in Experiment I-II using the same TMS coil, though still within approximately one standard deviation. This is likely due to the use of a gel pad, which introduces more distance between the TMS coil and the scalp, thus requiring a higher TMS intensity to evoke the same motor activity.”

Regarding the EMG gain, this did not affect TMS intensities and did not impact the measured neuromodulatory effects. The EMG gain at acquisition is always considered during signal digitization and further analyses.

(1.18) Exp. 4. It would be interesting to provide the changes in MEP amplitudes for those subjects who rated "inaudible" in the self-rating compared to the others. That's an important part of the interpretation: inaudible conditions lead to inhibition, so there is an effect. The auditory confound is not additive to the TUS effect.

Previously, we only provided participant’s ratings of audibility, and showed that conditions that were rated as inaudible more often showed less inhibition, descriptively indicating that inaudible stimulation does not lead to inhibition. This interpretation is in line with our conclusion that the TUS auditory confound acts as a cue signaling the upcoming TMS pulse, thus leading to preparatory inhibition.

We have now included an additional plot and discussion in Supplementary Figure 8 (Subjective Report of TUS Audibility). Here, we show the change in MEP amplitude from baseline for the three continuously masked TUS intensities as in the main manuscript, but now split by participant rating of audibility. Descriptively, less audible sounds result in no marked change or a smaller change in MEP amplitude. This supports our conclusion that direct neuromodulation is not being observed here. When participants were unsure whether they could hear TUS, or when they did hear TUS, more inhibition was observed. However, this is still to a lesser degree than unmasked stimulation which was nearly always audible, and likely also more salient. This also supports our conclusion that these results indicate a role of cue salience rather than direct neuromodulation. Regarding masked conditions where participants were uncertain whether they heard TUS, the sound was likely sufficient to act as a cue, albeit potentially subliminally. After all, preparatory inhibition is not a conscious action undertaken by the participant either. We would also like to note that participants reported perceived audibility after each block, not after each trial, so selfreported audibility was not a fine-grained measurement. The data from Experiment IV suggest that the volume of the cue has an impact on motor inhibition. Taken together with the points mentioned in 1.16, it is not possible to conclude there is evidence for direct neuromodulation in Experiment IV.

(1.19) I suggest to re-order sub panels of the main figures to fit with the chronologic order of appearance in the text. (e.g Figure 1 with (A) Ultrasonic parameters, (B) 3D-printed clamp, (C) Sound-TMS coupling, (D) Experimental condition).

We have restructured the figures in the manuscript to provide more clarity and to have greater alignment with the eLife format.

(2.1) Although auditory confounds during TUS have been demonstrated before, the thorough design of the study will lead to a strong impact in the field.

We thank the reviewer for recognition of the impact of our work. They highlight that auditory confounds during TUS have been demonstrated previously. Indeed, our work builds upon a larger research line on auditory confounds. The current study extends on the confound’s presence by quantifying its impact on motor cortical excitability, but perhaps more importantly by invalidating the most robust and previously replicable findings in humans. Further, this study provides a way forward for the field, highlighting the necessity of (in)active control conditions and tightly matched sham conditions for appropriate inferences in future work. We have amended the abstract to better reflect these points:

“Primarily, this study highlights the substantial shortcomings in accounting for the auditory confound in prior TUS-TMS work where only a flip-over sham control was used. The field must critically reevaluate previous findings given the demonstrated impact of peripheral confounds. Further, rigorous experimental design via (in)active control conditions is required to make substantiated claims in future TUS studies.”

(2.2) A few minor [weaknesses] are that (1) the overview of previous related work, and how frequent audible TUS protocols are in the field, could be a bit clearer/more detailed

We have expanded on previous related work in the revised manuscript:

“Indeed, there is longstanding knowledge of the auditory confound accompanying pulsed TUS (Gavrilov & Tsirulnikov, 2012). However, this confound has only recently garnered attention, prompted by a pair of rodent studies demonstrating indirect auditory activation induced by TUS (Guo et al., 2022; Sato et al., 2018). Similar effects have been observed in humans, where exclusively auditory effects were captured with EEG measures (Braun et al., 2020). These findings are particularly impactful given that nearly all TUS studies employ pulsed protocols, from which the pervasive auditory confound emerges (Johnstone et al., 2021).”

(2.3) The acoustic control stimulus can be described in more detail

We have elaborated upon the masking stimulus for each experiment in the revised manuscript as follows:

Experiment I: “In addition, we also included a sound-only sham condition that resembled the auditory confound. Specifically, we generated a 1000 Hz square wave tone with 0.3 ms long pulses using MATLAB. We then added white noise at a signal-to-noise ratio of 14:1. This stimulus was administered to the participant via bone-conducting headphones.”

Experiment II: “In this experiment, the same 1000 Hz square wave auditory stimulus was used for sound-only sham and auditory masking conditions. This stimulus was administered to the participant over in-ear headphones.”

Experiment III: “Auditory stimuli were either 500 or 700 ms in duration, the latter beginning 100 ms prior to TUS (Supplementary Fig. 3.3). Both durations were presented at two pitches. Using a signal generator (Agilent 33220A, Keysight Technologies), a 12 kHz sine wave tone was administered over speakers positioned to the left of the participant as in Fomenko and colleagues (2020). Additionally, a 1 kHz square wave tone with 0.5 ms long pulses was administered as in Experiments I, II, IV, and prior research (Braun et al., 2020) over noisecancelling earbuds.”

Experiment IV: “We additionally applied stimulation both with and without a continuous auditory masking stimulus that sounded similar to the auditory confound. The stimulus consisted of a 1 kHz square wave with 0.3 ms long pulses. This stimulus was presented through wired bone-conducting headphones (LBYSK Wired Bone ConductionHeadphones). The volume and signal-to-noise ratio of the masking stimulus were increased until the participant could no longer hear TUS, or until the volume became uncomfortable.”

In the revised manuscript we have also open-sourced the audio files used in Experiments I, II, and IV, as well as a recording of the output of the signal generator for Experiment III:

“Auditory stimuli used for sound-sham and/or masking for each experiment are accessible here: https://doi.org/10.5281/zenodo.8374148.”

(2.4) The finding that remaining motor inhibition is observed during acoustically masked trials deserves further discussion.

We agree. Please refer to points 1.16 and 1.18.

(2.5) In several places, the authors state to have "improved" control conditions, yet remain somewhat vague on the kind of controls previous work has used (apart from one paragraph where a similar control site is described). It would be useful to include more details on this specific difference to previous work.

In the revised manuscript, we have clarified the control condition used in prior studies as follows:

Abstract:

“Primarily, this study highlights the substantial shortcomings in accounting for the auditory confound in prior TUS-TMS work where only a flip-over sham control was used.”

Introduction:

“To this end, we substantially improved upon prior TUS-TMS studies implementing solely flip-over sham by including both (in)active control and multiple sound-sham conditions.”

Methods:

“We introduced controls that improve upon the sole use of flip-over sham conditions used in prior work. First, we applied active control TUS to the right-hemispheric face motor area, allowing for the assessment of spatially specific effects while also better mimicking ontarget peripheral confounds. In addition, we also included a sound-only sham condition that closely resembled the auditory confound.”

(2.6) I also wondered how common TUS protocols are that rely on audible frequencies. If they are common, why do the authors think this confound is still relatively unexplored (this is a question out of curiosity). More details on these points might make the paper a bit more accessible to TUS-inexperienced readers.

Regarding the prevalence of the auditory confound, please refer to point 2.2.

Peripheral confounds associated with brain stimulation can have a strong impact on outcome measures, often even overshadowing the intended primary effects. This is well known from electromagnetic stimulation. For example, the click of a TMS pulse can strongly modulate reaction times (Duecker et al., 2013, PlosOne) with effect sizes far beyond that of direct neuromodulation. Unfortunately, this consideration has not yet fully been embraced by the ultrasonic neuromodulation community. This is despite long known auditory effects of TUS (Gavrilov & Tsirulnikov, 2012, Acoustical Physics). It was not until the auditory confound was shown to impact brain activity by Guo et al., and Sato et al., (2018, Neuron) that the field began to attend to this phenomenon. Mohammadjavadi et al., (2019, BrainStim) then showed that neuromodulation persisted even in deaf mice, and importantly, also demonstrated that ramping ultrasound pulses could reduce the auditory brainstem response (ABR). Braun and colleagues (2020, BrainStim) were the first bring attention to the auditory confound in humans, while also discussing masking stimuli. This was followed by a study from Johnstone and colleagues (2021, BrainStim) who did preliminary work assessing both masking and ramping in humans. Recently, Liang et al., (2023) proposed a new form of masking colourfully titled the ‘auditory Mondrian’. Further research into the peripheral confounds associated with TUS is on the way.

However, we agree that the confound remains relatively unexplored, particularly given the substantial impact it can have, as demonstrated in this paper. What is currently lacking is an assessment of the reproducibility of previous work that did not sufficiently consider the auditory confound. The current study constitutes a strong first step to addressing this issue, and indeed shows that results are not reproducible when using control conditions that are superior to flip-over sham, like (in)active control conditions and tightly matched soundsham conditions. This is particularly important given the fundamental nature of this research line, where TUS-TMS studies have played a central role in informing choices for stimulation protocols in subsequent research.

We would speculate that, with TUS opening new frontiers for neuroscientific research, there comes a rush of enthusiasm wherein laying the groundwork for a solid foundation in the field can sometimes be overlooked. Therefore, we hope that this work sends a strong message to the field regarding how strong of an impact peripheral confounds can have, also in prior work. Indeed, at the current stage of the field, we see no justification not to include proper experimental control moving forward. Only when we can dissociate peripheral effects from direct neuromodulatory effects can our enthusiasm for the potential of TUS be warranted.

(2.7) Results, Fig. 2: Why did the authors not directly contrast target TUS and control conditions?

Please refer to point 1.1.

(2.8) The authors observe no dose-response effects of TUS. Does increasing TUS intensity also increase an increase in TUS-produced sounds? If so, should this not also lead to doseresponse effects?

We thank the reviewer for this insightful question. Yes, increasing TUS intensity results in an increased volume of the auditory confound. Under certain circumstances this could lead to ‘dose-response’ effects. In the manuscript, we propose that the auditory confounds acts as a cue for the upcoming TMS pulse, thus resulting in MEP attenuation once the cue is informative (i.e., when TMS timing can be predicted by the auditory confound). In this scenario, volume can be taken as the salience of the cue. When the auditory confound is sufficiently salient, it should cue the upcoming TMS pulse and thus result in a reduction of MEP amplitude.

If we take Experiment II as an example (Figure 3B), the 19.06 W/cm2 stimulation would be louder than the 6.35 W/cm2 intensity. However, as both intensities are audible, they both cue the upcoming TMS pulse. One could speculate that the very slight (nonsignificant) further decrease for 19.06 W/cm2 stimulation could owe to a more salient cueing.

One might notice that MEP attenuation is less strong in Experiment I, even though higher intensities were applied. Directly contrasting intensities from Experiments I and II was not feasible due to differences in transducers and experimental design. From the perspective of sound cueing of the upcoming TMS pulse, the auditory confound cue was less informative in Experiment I than Experiment II, because TUS stimulus durations of both 100 and 500 ms were administered, rather than solely 500 ms durations. This could explain why descriptively less MEP attenuation was observed in Experiment I, where cueing was less consistent.

Perhaps more convincing evidence of a sound-based ‘dose-response’ effect comes from Experiment IV (Figure 4B). Here, we propose that continuous masking reduced the salience of the auditory confound (cue), and thus, less MEP attenuation was be observed. Indeed, we see less MEP change for masked stimulation. For the lowest administered volume during masked stimulation, there was no change in MEP amplitude from baseline. For higher volumes, however, there was a significant inhibition of MEP amplitude, though it was still less attenuation than unmasked stimulation. These results indicate a ‘doseresponse’ effect of volume. When the volume (intensity) of the auditory confound was low enough, it was inaudible over the continuous mask (also as reported by participants), and thus it did not act as a cue for the upcoming TMS pulse, therefore not resulting in motor inhibition. When the volume (intensity) was higher, less participants reported not being able to hear the stimulation, so the cue was to a given extent more salient, and in line with the cueing hypothesis more inhibition was observed.

In summary, because the volume of the auditory confound scales with the intensity of TUS, there may be dose-response effects of the auditory confound volume. Along the border of (in)audibility of the confound, as in masked trials of Experiment IV, we may observe dose-response effects. However, at clearly audible intensities (e.g., Experiment I & II), the size of such an effect would likely be small, as both volumes are sufficiently audible to act as a cue for the upcoming TMS pulse leading to preparatory inhibition.

(2.9) I wonder if the authors could say a bit more on the acoustic control stimulus. Some sound examples would be useful. The authors control for audibility, but does the control sound resemble the one produced by TUS?

Please refer to point 2.3.

(2.10) The authors' claim that the remaining motor inhibition observed during masked trials is due to persistent audibility of TUS relies "only" on participants' descriptions. I think this deserves a bit more discussion. Could this be evidence that there is a TUS effect in addition to the sound effect?

Please refer to points 1.16 and 1.18.